# Induction of memory-like dendritic cell responses in vivo

Camaron R. Hole[1,2], Chrissy M. Leopold Wager[1,2], Natalia Castro-Lopez[1,2], Althea Campuzano[1,2], Hong Cai[1,2], Karen L. Wozniak[1,2], Yufeng Wang[1,2] & Floyd L. Wormley Jr.[1,2]

Dendritic cells (DCs), a vital component of the innate immune system, are considered to lack antigen specificity and be devoid of immunological memory. Strategies that can induce memory-like responses from innate cells can be utilized to elicit protective immunity in immune deficient persons. Here we utilize an experimental immunization strategy to modulate DC inflammatory and memory-like responses against an opportunistic fungal pathogen that causes significant disease in immunocompromised individuals. Our results show that DCs isolated from protectively immunized mice exhibit enhanced transcriptional activation of interferon and immune signaling pathways. We also show long-term memory-like cytokine responses upon subsequent challenge with the fungal pathogen that are abrogated with inhibitors of specific histone modifications. Altogether, our study demonstrates that immunization strategies can be designed to elicit memory-like DC responses against infectious disease.

[1] Department of Biology, The University of Texas at San Antonio, San Antonio, TX 78249, USA. [2] The South Texas Center for Emerging Infectious Diseases, The University of Texas at San Antonio, San Antonio, TX 78249, USA. Correspondence and requests for materials should be addressed to F.L.W. Jr. (email: floyd.wormley@utsa.edu)

Dendritic cells (DCs) are cells of the innate immune system that bridge innate and adaptive immune responses and are essential mediators of immunity and tolerance[1]. DCs, macrophages, and B cells are considered professional antigen presenting cells (APCs), but only DCs are able to present antigens to naïve T cells[2]. Immature DCs are constantly sampling the environment for antigens, which they take up by endocytosis, phagocytosis, or pinocytosis. Once antigen is taken up, the DCs undergo maturation and up-regulate co-stimulatory molecules that are required for complete T cell activation.

While DCs have historically been investigated for their potent ability to activate the adaptive immune system, DCs also play an important role in shaping and directing the innate immune response[3]. As DCs are among the first cells to detect an invading pathogen and respond, DCs can produce copious amounts of cytokines that can activate and otherwise direct the activity of other innate immune cells while they instruct T cells to further refine the overall immune response[4–6]. For example, CD8α[+] DCs were identified as the cellular source of IL-12 during *Toxoplasma gondii* infection and ablation of these DCs rendered mice extremely susceptible to *T. gondii*[4].

Cells of the monocytic lineage have the ability to polarize toward different activation phenotypes depending upon the specific stimuli and cytokine environment[7,8]. Studies demonstrate that DCs, like macrophages, can be induced to preferentially drive pro-inflammatory CD4[+] T helper1 (Th1) type or anti-inflammatory CD4[+] Th2 type immune responses. For example, DCs can be conditioned during allergy and helminth infections to prime Th2 type responses[9,10]. Additionally, treatment of DCs with IL-4 induced expression of multiple alternative activation markers, although with a different expression pattern compared to that exhibited by macrophages[11]. IL-4 induces robust expression of resistin-like molecule alpha (RELMα) also known as FIZZ1. DCs that induce Th1 type responses produce IL-12 and reactive nitrogen species and switch their metabolism from mitochondrial oxidative phosphorylation to glycolysis through the activity of the mammalian target of rapamycin (mTOR)[12–16] in a similar fashion to M1 macrophages. While macrophage activation has been extensively studied, DC activation phenotype remains largely unknown.

Traditionally, DCs are considered to lack antigen-specificity and not possess immunological memory; characteristics long believed to only rest with T and B cells. However, studies have clearly shown evidence for innate immune memory by monocytes, macrophages, and NK cells (reviewed in[17]). To date, there is little definitive evidence that DCs can be induced to elicit long-term enhanced immunological responses against subsequent exposure to a pathogen. To address this gap in knowledge, we utilized a protective fungal vaccine model to study DC polarization and long-term innate memory responses against the opportunistic fungal pathogen *Cryptococcus neoformans*. In the current study, we observed that DCs activated during the immunization phase within protectively immunized mice exhibited a phenotype similar to M1 activated macrophages. In contrast, DCs isolated from naïve mice infected with wild-type *C. neoformans* exhibited a phenotype similar to M2 activated macrophages. During the anamnestic response, DCs from the protectively immunized mice displayed a predominantly pro-inflammatory phenotype, increased interferon response gene expression, and enhanced *Cryptococcus*-specific cytokine recall responses upon subsequent fungal challenge that were abrogated in the presence of specific histone modification inhibitors. Overall, these studies demonstrate that immunization strategies can be designed to elicit long-lasting memory-like DC responses to modulate host immune responses toward the resolution of infectious disease.

## Results

**Fungal infection results in differential DC phenotypes.** *Cryptococcus neoformans* is a common cause of pneumonia in immune compromised individuals and has a unique predilection to disseminate from the lung to the central nervous system (CNS) resulting in life-threatening meningoencephalitis[18,19]. DC polarization during pulmonary *Cryptococcus* infection has largely gone unstudied. Consequently, we chose to determine DC polarization profiles using a pulmonary fungal vaccine model system previously employed to study M1 and M2 polarization phenotypes[20,21]. Mice given an experimental pulmonary infection with a *C. neoformans* strain engineered to produce murine interferon-γ (IFN-γ), denoted H99γ, respond with strong Th1 type immune responses, M1 activation and resolution of the infection. Alternatively, infection with the parental wild-type (WT) *C. neoformans* strain H99 typically elicits strong Th2 type immune responses, M2 activation and progressive disease leading to complete mortality in mice[22–24]. Herein, we demonstrate that mice given an experimental pulmonary infection with either *C. neoformans* strains H99γ or H99 have robust recruitment of both CD103[+] conventional DC (cDC) and CD11b[+] cDC subsets (Fig. 1a, b). Nonetheless, we observed a significant increase in the absolute number of CD11b[+] cDCs in the lungs of H99γ infected mice at day 7 post-inoculation compared to mice given WT cryptococci (Fig. 1b). The increase in CD11b[+] cDCs at day 7 post-inoculation occurred while the fungal burdens in the lung, brain and spleen of H99 and H99γ infected mice were at similar levels (Fig. 1d–f). Weight loss, an indicator of disease exacerbation, was also similar in both groups of mice (Fig. 1c) although one WT infected mouse succumbed to infection on day 13 post-infection and thus were not included in the weight loss analysis for the remainder of the study. Mortality due to pulmonary infection with *C. neoformans* strain H99 is oftentimes due to asphyxiation. Indeed, we began to observe labored breathing in WT infected, but not the H99γ infected, mice towards the conclusion of the observation period. We observed similar levels of CD103[+] cDCs and CD11b[+] cDCs in both groups at days 14 and 20 post-infection (Fig. 1a, b). However, the fungal burden continued to increase in WT infected mice whereas H99γ infected mice had either resolved or appeared to be in the process of resolving the infection (Fig. 1d–f). We observed that both the absolute number of yeast detected in pulmonary, brain and splenic tissues and the percentage of tissues with detectable yeast were significantly less in H99γ infected mice compared to that observed in mice infected with WT yeast. Dissemination of yeasts to the brain and splenic tissues of H99 and H99γ infected mice occurred. Nonetheless, the presence of H99γ in brain and splenic tissues appeared to decline to undetectable levels in the majority of mice analyzed at day 20 post-infection whereas fungal burden significantly increased in H99 infected mice during the observation period.

Since studies show that DCs can be induced to preferentially prime Th1 or Th2 type responses and express markers typically utilized to determine M1 macrophage activation (IFN-γ, IL-17a, TNF-α, NOS2, CXCL9, and CXCL10) and M2 macrophage activation (Arg1, FIZZ1, YM1, IL-4, and IL-13) phenotypes (Fig. 2a)[9–11], we next sought to determine the activation status of the DCs during H99 and H99γ infection. We observed significantly increased transcript levels of NOS2, CXCL9, and CXCL10 beginning at day 5 post-infection in DCs isolated from H99γ infected mice compared to DCs from H99 infected mice (Fig. 2a). Some transcript levels (IFN-γ, TNF-α, and IL-17a) remained elevated in H99γ infected mice compared to H99 infected mice during a time when pulmonary fungal burden is controlled (day 14; Fig. 1d) while other transcripts (NOS2, CXCL9 and CXCL10) were no longer elevated. This observation

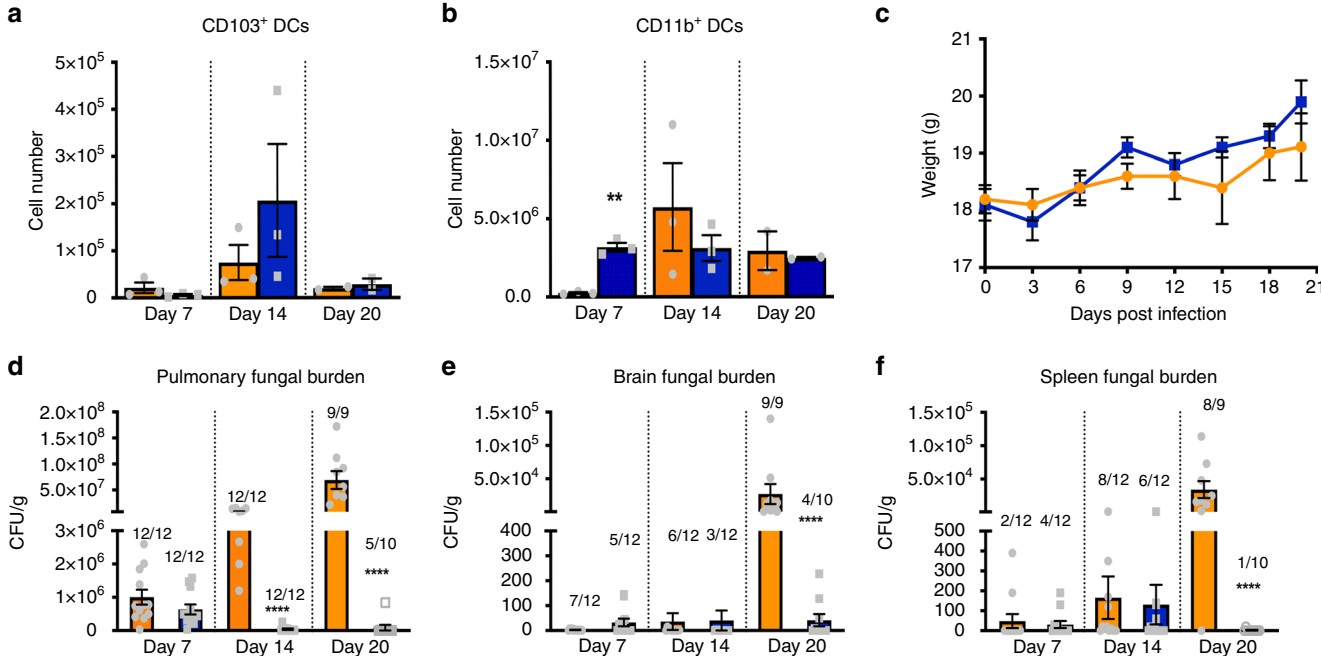

**Fig. 1** Pulmonary infection with *C. neoformans* strain H99γ accelerates cDC1 response. BALB/c mice were inoculated via intranasal inhalation with $10^4$ CFU *C. neoformans* strain H99γ or H99. Pulmonary leukocytes were then isolated from the right lobes of enzymatically digested lungs on days 7, 14, and 20 post-inoculation and analyzed by flow cytometry. Individual mice were weighed every 72 h until conclusion of the observation period (day 20 post-inoculation). Pulmonary, brain, and spleen fungal burden was quantified from tissue homogenates on days 7, 14, and 20 post-inoculation. **a** Absolute numbers of $CD103^+$ conventional (c) DCs ($CD11c^+/CD24^+/CD11b^-/CD103^+$), and **b** $CD11b^+cDCs$ ($CD11c^+/CD24^+/CD11b^+/CD172\alpha^+$) in the lung. Data shown are from three experiments for days 7 and 14 and two experiments for day 20 with five mice per group. **c** Data shown are from 10 mice per group. **d–f** Pulmonary, brain, and spleen fungal burden data shown are from three experiments with 4–5 mice per group and results expressed as mean log CFU per gram of tissue. Bars indicate the mean ± standard error of the mean (SEM). (\**p* < 0.05, \*\**p* < 0.01, \*\*\**p* < 0.001); unpaired Student's *t*-test (two-tailed)

suggests that the immune response is receding in parallel with the reduction in pulmonary fungal burden. We further observed significant decreases in transcripts of Arg1, FIZZ1, YM1, IL-4, and IL-13 in DCs from H99γ infected mice compared to DCs from H99 infected mice throughout most selected time points. (Fig. 2b). These results strongly suggest that the protective immune response to *C. neoformans* is associated with a phenotype similar to that observed with M1 polarized macrophages while non-protective responses are associated with a DC phenotype akin to that observed by M2 polarized macrophages in the lung. The NOS2/Arg1 expression ratio is used to specify M1/M2 polarized macrophages with a higher NOS2 to Arg1 expression ratio indicating an M1 polarized phenotype, and a lower NOS2 to Arg1 expression ratio indicating an M2 polarized phenotype[22]. In the current study, we observed a higher NOS2 to Arg1 expression ratio in DCs isolated from H99γ infected mice at all time points compared to DCs from H99 infected mice (Fig. 2c). We observed a skewing in DCs isolated from H99γ infected mice at all time points compared to DCs from H99 infected mice when we analyzed the CXCL9/FIZZ1 expression ratio (Fig. 2d). However, while the NOS2/Arg1 expression ratio remained constantly elevated in DCs isolated from H99γ infected mice at all time points, the CXCL9/FIZZ1 expression ratio peaks at day 7 post-inoculation and declines by day 14, further indicating the infection is being controlled. Collectively, these data indicate that infection with *C. neoformans* strain H99γ and H99 results in differential polarization of DC activation phenotypes.

**Protective vaccination induces pro-inflammatory DC profile.** Immunization with *C. neoformans* strain H99γ results in the induction of sterile immunity against subsequent challenge with a

WT *C. neoformans* strain H99 that does not produce IFN-γ whereas mice immunized with heat-killed *C. neoformans* strain H99γ (HKH99γ) yeast succumb to subsequent challenge[21,25,26]. Therefore, we next evaluated DC polarization during protective and non-protective immune responses against challenge with WT *Cryptococcus* yeast in mice immunized with live *C. neoformans* strain H99γ or HKH99γ, respectively. Since the kinetics of pulmonary DC recruitment and fungal clearance is significantly accelerated in mice immunized with H99γ[27], our analyses were focused on much earlier time points. We observed that mice immunized with either *C. neoformans* strain H99γ or HKH99γ had equivalent percentages of $CD103^+$ cDC (2.65 ± 0.52 SEM and 1.32 ± 0.34 SEM in *C. neoformans* strain H99γ or HKH99γ, respectively, immunized mice, unpaired Student's *t*-test (two-tailed)) or $CD11b^+$ cDC (12.53 ± 2.56 SEM and 11.13 ± 3.24 SEM in *C. neoformans* strain H99γ or HKH99γ, respectively, immunized mice, unpaired Student's *t*-test (two-tailed)) subsets in their lungs at day 1 post-challenge. Additional analysis of $CD11b^+$ cells indicated that less than 1% of this population in the lungs of *C. neoformans* strain H99γ and HKH99γ immunized mice were monocyte-derived DCs ($CD11c^+/CD11b^+/CD24^-/CD64^+/Fc\epsilon R1\alpha^+/Ly6C^+/I-A/I-E^+$; Supplementary Fig. 1)[10,28,29].

DC polarization in the lungs of immunized mice were assessed on days 1 and 3 post-challenge with WT *C. neoformans* strain H99 as described above. We observed significantly increased transcript levels of NOS2 and CXCL9 on days 1 and day 3 post-challenge in protectively immunized mice compared to mice immunized with HKH99γ (Fig. 3a). Also, we observed a significant decrease in markers FIZZ1 on days 1 and day 3 and CD206 on day 3 post-challenge in protectively immunized mice (Fig. 3b). However, there was also a significant increase in Arg1 transcript levels on day 3 post-challenge within DCs of H99γ

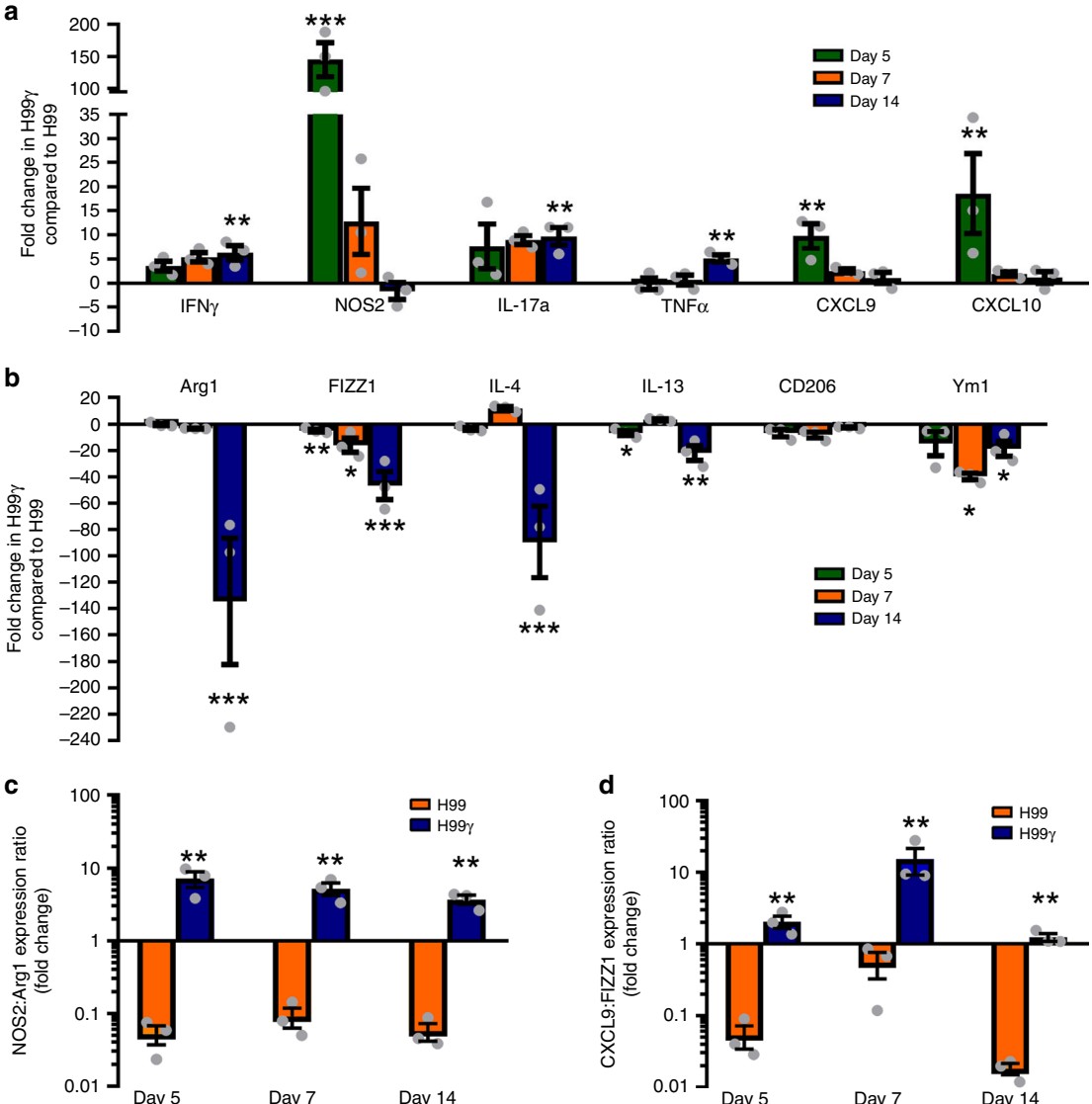

**Fig. 2** *C. neoformans* strain H99 or H99γ infection results in differential DC phenotypes. BALB/c mice were inoculated via intranasal inhalation with $10^4$ CFU *C. neoformans* strain H99γ or H99. **a**, **b** Pulmonary leukocytes were isolated on days 5, 7, and 14 post-inoculation and depleted of CD3+ cells by positive selection using α-CD3e labeled magnetic beads and macrophages by positive selection using biotinylated α-F4/80 and α-biotin magnetic beads. Real-time PCR analysis of each transcript was normalized to GAPDH. Bars represent the log10-fold change in gene expression during infection with *C. neoformans* strain H99γ compared to wild-type *C. neoformans* strain H99. **c** NOS2:Arg1 expression ratio. **d** CXCL9:FIZZ1 expression ratio. Data shown are cumulative of three independent experiments utilizing pooled DCs from five mice per group per experiment. Bars indicate the mean ± standard error of the mean (SEM). (*$p < 0.05$, **$p < 0.01$, ***$p < 0.001$); unpaired Student's *t*-test (two-tailed)

immunized mice compared to mice immunized with HKH99γ (Fig. 3b), consistent with an overall increased pro-inflammatory DC profile but also some, albeit fewer, elements of anti-inflammatory DCs emerging as the infection is beginning to be controlled due to the protective anamnestic anti-*Cryptococcus* response[21]. Additionally, we observed a significant increase in the NOS2/Arg1 expression ratio in DCs isolated from H99γ immunized mice at day 1 post-challenge compared to DCs from HKH99γ immunized mice (Fig. 3c). The NOS2/Arg1 expression ratio appears to be similar in H99γ immunized and HKH99γ immunized mice at day 3 post-challenge (Fig. 3c). However, the leveling in the NOS2/Arg1 expression ratio in H99γ immunized at day 3 post-challenge appears to be due to a high but equivalent induction of NOS2 and Arg1 levels relative to that observed in HKH99γ immunized mice and not a reduction in overall NOS2 and/or Arg1 expression (Fig. 3a, b).

In contrast, the CXCL9/FIZZ1 expression ratio continues to increase at all time points in H99γ immunized mice (Fig. 3d). Collectively, these data demonstrate that protective recall responses in H99γ immunized mice challenged with WT *C. neoformans* yeast resemble the pro-inflammatory DC profile observed during the protective immune response to acute infection with H99γ.

**Enhanced inflammatory transcripts in DCs of protected mice.** We next performed whole transcriptome analysis of DCs isolated from protectively compared to non-protectively immunized mice challenged with WT *C. neoformans* yeast. BALB/c mice were immunized with HKH99γ or H99γ and allowed to rest for 70 days. The immunized mice were then challenged with WT *C. neoformans* or left unchallenged to serve as controls for these

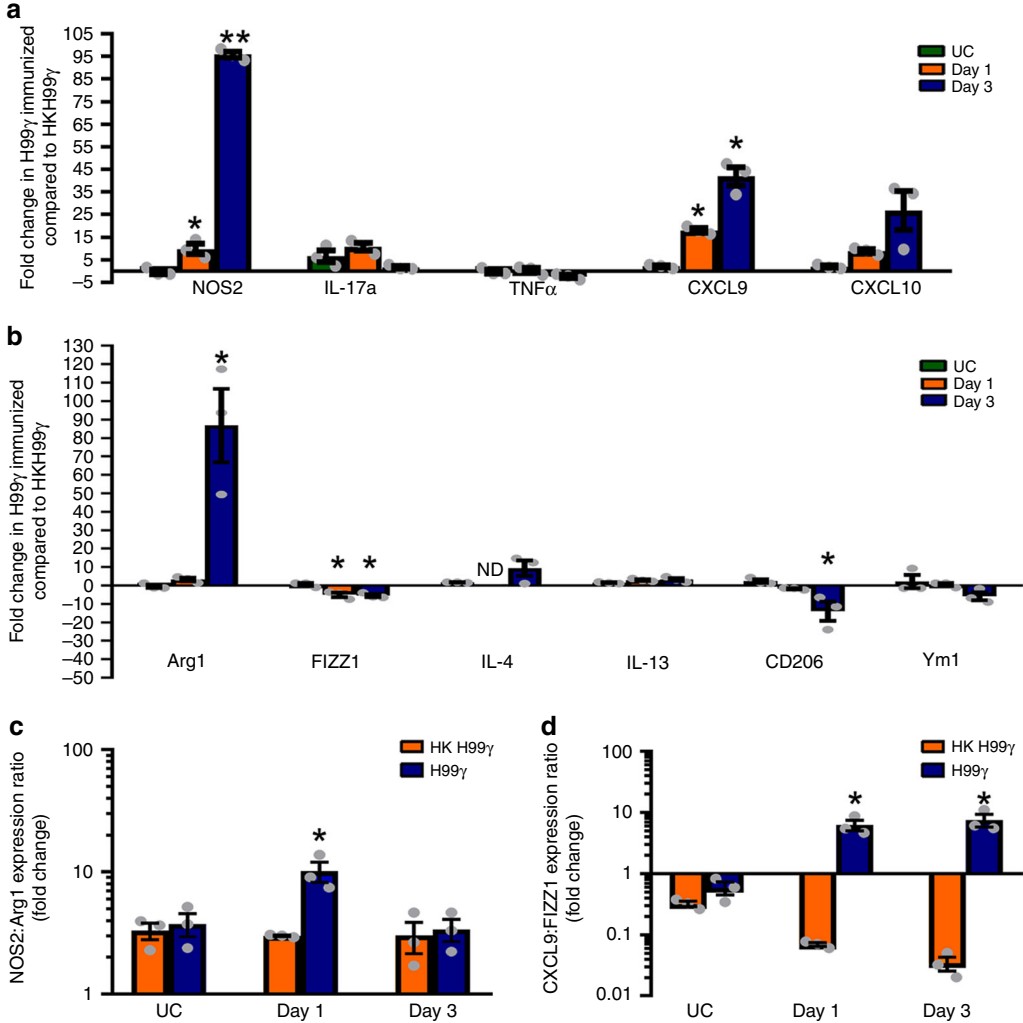

**Fig. 3** Protective immunization induces pro-inflammatory DC phenotype during challenge. BALB/c mice were intranasally immunized with $10^4$ CFU of *C. neoformans* strain H99γ or HKH99γ in 50 µl of sterile PBS. Seventy days later, mice were left unchallenged (UC) or subsequently infected with $10^4$ CFU of *C. neoformans* strain H99. Pulmonary leukocytes were isolated from lung tissues by enzymatic digestion on days 1 and 3 post-inoculation or from immunized but unchallenged mice. Leukocyte populations were depleted of CD3$^+$ cells by positive selection using α-CD3e labeled magnetic beads and macrophages by positive selection using biotinylated α-F4/80 and α-biotin magnetic beads. DCs were isolated by positive selection using CD11c$^+$ magnetic beads. **a**, **b** Real-time PCR analysis of each transcript was normalized to GAPDH. Bars represent the $\log_{10}$-fold change in gene expression in DCs from H99γ-immunized mice compared to non-protected mice. **c** NOS2:Arg1 expression ratio. **d** CXCL9:FIZZ1 expression ratio. Data shown are cumulative of three independent experiments utilizing pooled DCs from 20–25 mice per group per experiment and bars indicate the mean ± standard error of the mean (SEM). (*$p < 0.05$, **$p < 0.001$); unpaired Student's *t*-test (two-tailed)

studies. DCs were then isolated, as described above, from immunized but unchallenged mice and from immunized and challenged mice at day 1 post-challenge and total RNA extracted for whole transcriptome expression analysis. DCs isolated from H99γ immunized mice on day 1 post-challenge had 120 gene transcripts expressed at significantly higher levels compared to gene transcripts expressed in DCs from HKH99γ immunized mice (Fig. 4a). Alternatively, DCs isolated from HKH99γ immunized mice on day 1 post-challenge had six gene transcripts expressed at significantly higher levels compared to gene transcripts expressed in DCs from H99γ immunized mice (Fig. 4a). Network analysis revealed that the top canonical pathways expressed in DCs of protectively, H99γ, immunized mice on day 1 post-challenge included those associated with interferon signaling, cytokines with roles in mediating communication between immune cells, pathways facilitating communication between innate and adaptive immune cells (Fig. 4a). Whole transcriptome analysis at day 1 post-challenge also revealed upregulation of a

network of genes that center on STAT1 and genes downstream in the STAT1 pathway (Fig. 4b). The 12 highest expressed transcripts found in DCs from H99γ immunized mice at day 1 post-challenge include NOS2, CXCL9, CXCL10 and CXCL11, four interferon gamma induced GTPases (Igtp) and Ubiquitin D (Ubd), which are associated with DC maturation (Fig. 4c). In agreement with the above RT-PCR data, we observed significantly increased transcripts of NOS2, CXCL9, and CXCL10 in DCs isolated from the protectively immunized mice on day 1 post-challenge (Fig. 4c). Gene ontology (GO) analysis revealed GO terms associated with cellular responses to interferon beta and gamma, and defense response to protozoan and virus, and immune system process (Fig. 4d). Whole transcriptome analysis conducted at day 3 post-challenge also revealed upregulation of a network of genes that center on the STAT1 pathway and enhanced cytokine and inflammatory responses (Supplementary Fig. 2). Altogether, the phenotype of the protective recall response exhibited by DCs from H99γ immunized mice challenged with

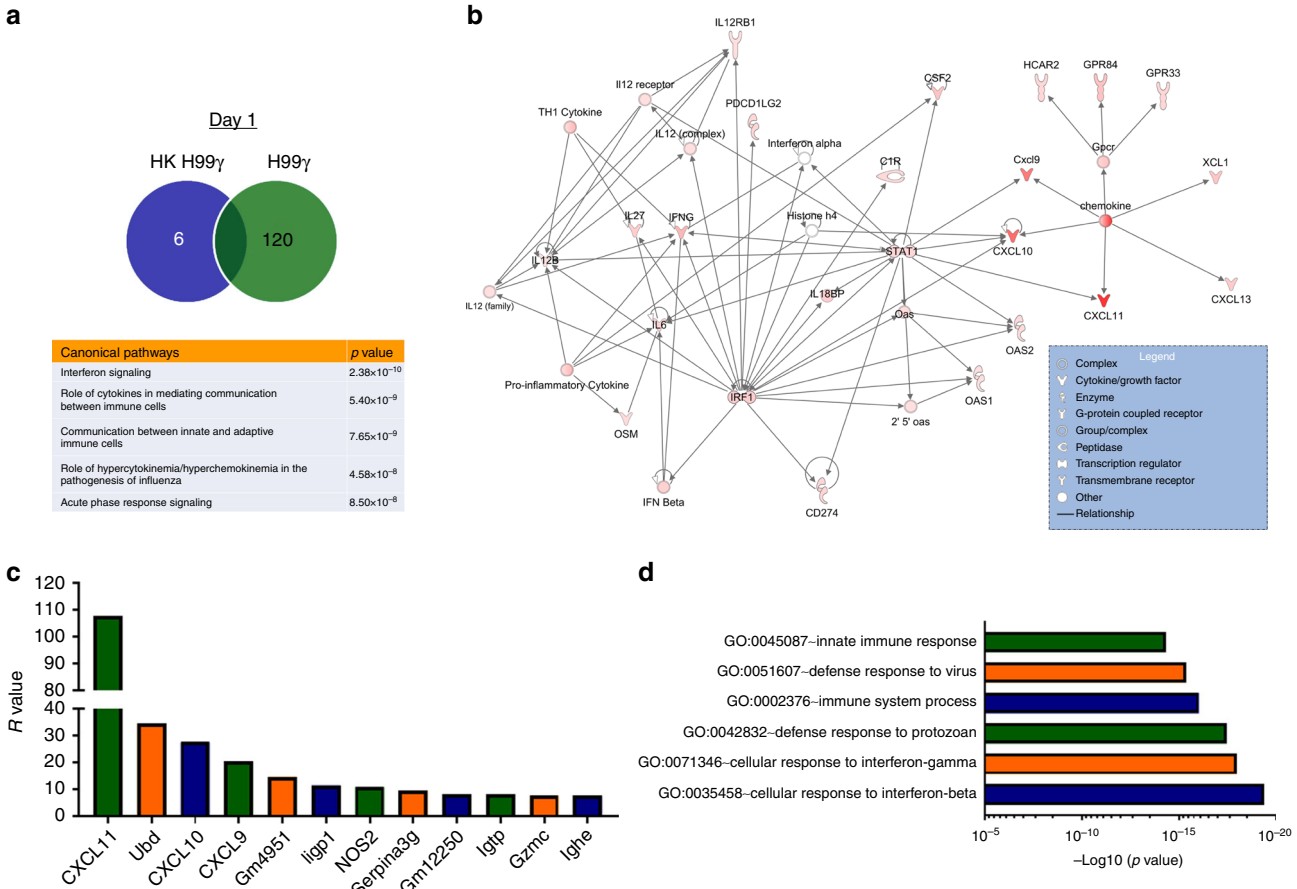

**Fig. 4** Enhanced interferon gamma response networks in DCs from protectively immunized mice. BALB/c mice were immunized via intranasal inhalation with $10^4$ CFU of *C. neoformans* strain H99γ or HKH99γ in 50 μl of sterile PBS. Seventy days later, mice were challenged with $10^4$ CFU of *C. neoformans* strain H99 via intranasal inhalation. Pulmonary leukocytes were isolated from lung tissues by enzymatic digestion on day 1 post-challenge. Leukocyte populations were depleted of $CD3^+$ cells by positive selection using α-CD3e labeled magnetic beads and macrophages by positive selection using biotinylated α-F4/80 and α-biotin magnetic beads. DCs were isolated by positive selection using $CD11c^+$ magnetic beads. Total mRNA from isolated DC populations was extracted and RNA-seq was performed. **a** Differently expressed genes and top canonical pathways and networks predicted by the Ingenuity Pathway Analysis software protectively immunized mice compared to non-protectively immunized mice as ranked by *p*-value. **b** Top networks up-regulated in day 1 post-challenge pulmonary DCs from protectively immunized mice compared to non-protectively immunized mice. The red color in the network figure corresponds to the log2 of expression fold change and represents increased gene expression in DCs from protectively compared to non-protectively immunized mice. **c** Fold change values for top 12 up-regulated genes from H99γ immunized DCs day 1 post-challenge. **d** Top six GO terms from H99γ immunized DCs day 1 post-challenge. Data were generated from a merged data set from three independent experiments with 20–25 mice per group

WT *Cryptococcus* yeast (H99) indicates that DCs from protectively immunized mice have a pro-inflammatory polarization profile.

**DCs from protectively immunized mice exhibit innate memory.** Previous studies in our lab showed that mice immunized with the H99γ strain and then depleted of both $CD4^+$ and $CD8^+$ T cells prior to and during challenge with WT *Cryptococcus* have an 80% survival rate[26]. These results suggest that vaccine-mediated protection against *Cryptococcus* is not solely dependent on T cell responses and that cells of the innate immune system likely compensate for the absence of T cells. Results from our RNA-Seq studies suggested that DCs from protectively immunized mice exhibit a memory-like response and react to *C. neoformans* challenge by rapidly increasing the gene expression of transcripts associated with interferon responses. To test whether or not DCs from protectively immunized mice exhibit innate memory responses, we performed cytokine recall experiments. BALB/c mice were immunized with HKH99γ or H99γ and allowed to rest for 70 days; a time span that allows for numerous

turnover of the DC population, which have a relatively short lifespan (4–6 days under steady state conditions) (Fig. 5a)[16,30]. Subsequently, leukocytes were isolated from the spleens of naïve and immunized mice and depleted of $CD3^+$ and $F4/80^+$ cells prior to $CD11c^+$ DC selection. We elected to use splenic DCs rather than DCs isolated from lung tissues to significantly reduce the number of mice needed to obtain a sufficient quantity of DCs to perform these assays. Flow cytometry was used to demonstrate the near absence of T cells following enrichment (<1.5% TCRα/β positive cells; Supplementary Fig. 3). The enriched DC population from immunized, unchallenged mice consisted exclusively of $CD103^+$ cDC and $CD11b^+$ cDC subsets (Supplementary Fig. 4). We did not detect monocyte-derived DCs in the enriched DC population (Supplementary Fig. 4), which is not surprising as we observed that <1% of the DC population within the spleens of immunized, unchallenged mice were monocyte-derived DCs prior to DC enrichment (Supplementary Fig. 5). Monocyte-derived DCs are typically not observed in tissues under non-inflammatory conditions, which may account for the absence of monocyte-derived DCs in the spleens of immunized,

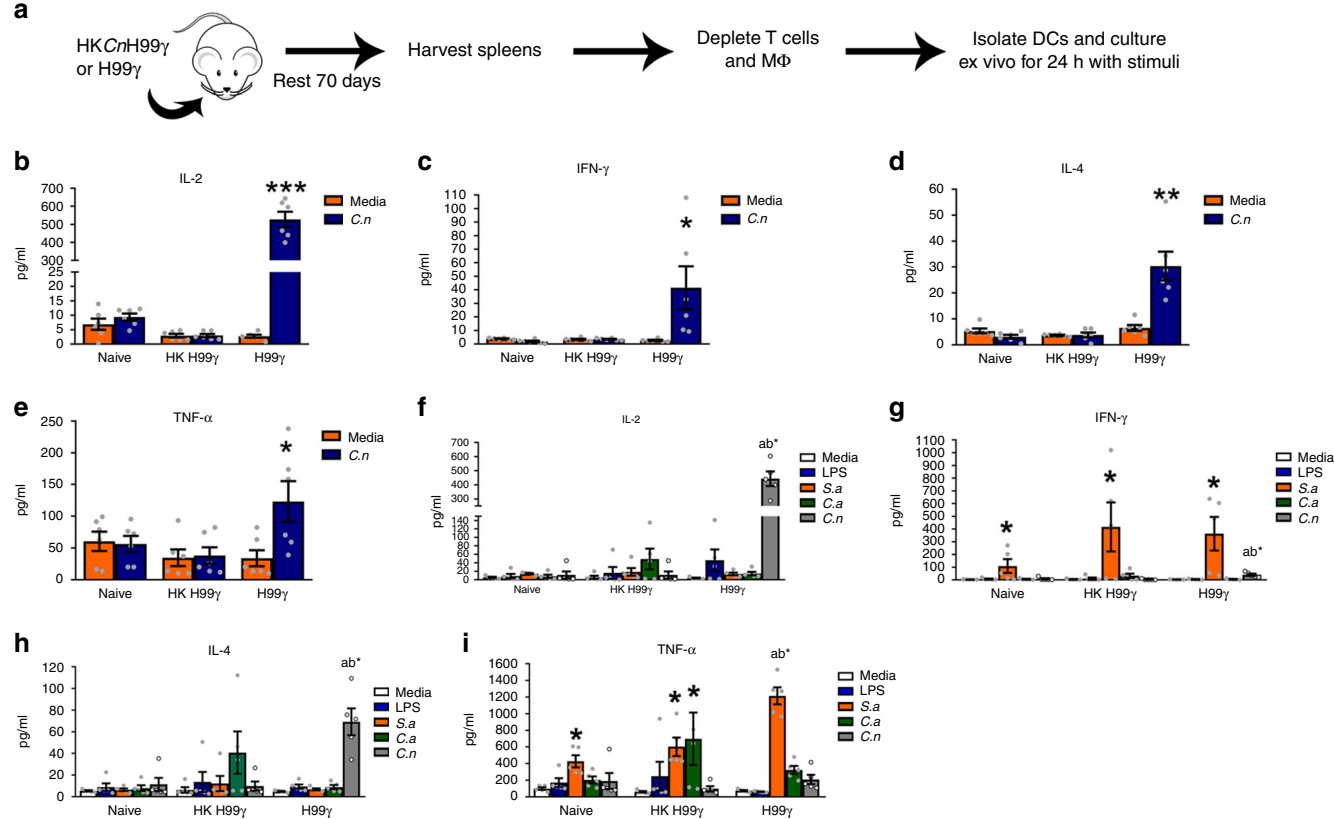

**Fig. 5** DCs from protectively immunized mice exhibit enhanced cytokine recall responses. **a** BALB/c mice were intranasally immunized with $10^4$ CFU of *C. neoformans* strain H99γ or HKH99γ in 50 μl of sterile PBS. After 70 days, DCs were isolated from the spleens of naïve, HKH99γ immunized or H99γ immunized mice after depletion of CD3+ and F4/80+ cells and stimulated with different antigens. **b–e** Isolated DCs were cultured alone or in the presence of a calcineurin alpha 1 subunit mutant (*cna1Δ*) of H99 in duplicate for 24 h after which supernatants were collected for cytokine analysis. **f–i** Isolated DCs were cultured in the presence of *C. neoformans cna1Δ*, LPS, heat-killed *Staphylococcus aureus*, or heat-killed *Candida albicans* yeast for 24 h and cytokine levels from the supernatants were analyzed. Results are cumulative of three experiments utilizing pooled DCs from 20 mice per group. Bars indicate the mean ± standard error of the mean (SEM). Symbols where significant differences were observed: a = compared to naïve, b = compared to HKH99γ, * = compared to media alone for that immunization group (*p < 0.05, **p < 0.01, ***p < 0.001); one-way ANOVA with the Tukey's multiple comparison test

unchallenged mice[10,31]. Isolated DCs were cultured alone or in the presence of a calcineurin alpha 1 subunit mutant (*cna1Δ*) of *C. neoformans* strain H99 which is unable to replicate at 37 °C. After 24 h, supernatants were collected for cytokine analysis. Significantly higher levels of IL-2, IFN-γ, IL-4, and TNF-α production were observed in cultures of DCs isolated from H99γ immunized mice exposed to *cna1Δ* compared to cultures of DCs from naïve or HKH99γ immunized mice exposed to *cna1Δ* (Fig. 5b–e). These results are significant in that they suggest that DCs from protectively immunized mice acquire long-lasting adaptive cell features that are maintained for several weeks following resolution of H99γ infection and are associated with trained immune responses against *C. neoformans*. We analyzed intracellular cytokine production by the DCs using imaging flow cytometry in order to confirm the source of the cytokines detected in culture. After 6 h in culture, we observed that DCs did indeed produce the cytokines IL-2, IL-4, IFN-γ, and TNF-α (Supplementary Fig. 6).

One of the hallmarks of trained immunity is that the training may provide cross-protection against other potential pathogens[32]. To test the specificity of the DC cytokine recall response demonstrated by DCs from H99γ immunized mice, DCs isolated from the spleens of naïve, HKH99γ or H99γ immunized mice were cultured in the presence of the *C. neoformans cna1Δ*, LPS, heat-killed *Staphylococcus aureus*, or heat-killed *Candida albicans*

yeast for 24 h and cytokine levels from the supernatants analyzed. We observed no significant differences in cytokine responses of DCs isolated from naïve or immunized mice exposed to LPS (Fig. 5f–i). In contrast, we observed significantly higher levels of IL-2, IFN-γ, and IL-4 cytokines following culture of DCs isolated from H99γ immunized mice exposed to *cna1Δ* compared to cultures of DCs from naïve or HKH99γ immunized mice exposed to *cna1Δ* (Fig. 5f–i). These results are suggestive of a *C. neoformans*-specific cytokine recall response by DCs isolated from protectively immunized mice. We also observed significant TNF-α responses to *S. aureus* by DCs isolated from naïve, HKH99γ, and H99γ immunized mice. (Fig. 5i). However, significantly more TNF-α was produced in response to *S. aureus* by DCs from H99γ immunized mice compared to DCs from naïve and HKH99γ immunized mice (Fig. 5i). Thus, DCs may be primed to elicit greater TNF-α responses to re-challenge with antigen in a non-specific manner suggesting that the recall response of primed DCs varies depending on the antigen and/or cytokine analyzed. Metabolic changes are also reported to accompany the trained immunity phenotype in monocytes and macrophages; particularly, a shift to aerobic glycolysis[33–35]. However, we observed no significant difference in lactate release, an indicator of aerobic glycolysis, between DCs from HKH99γ or H99γ immunized mice following culture in the presence of the *C. neoformans cna1Δ* for 24 h (Supplementary Fig. 7). Nonetheless,

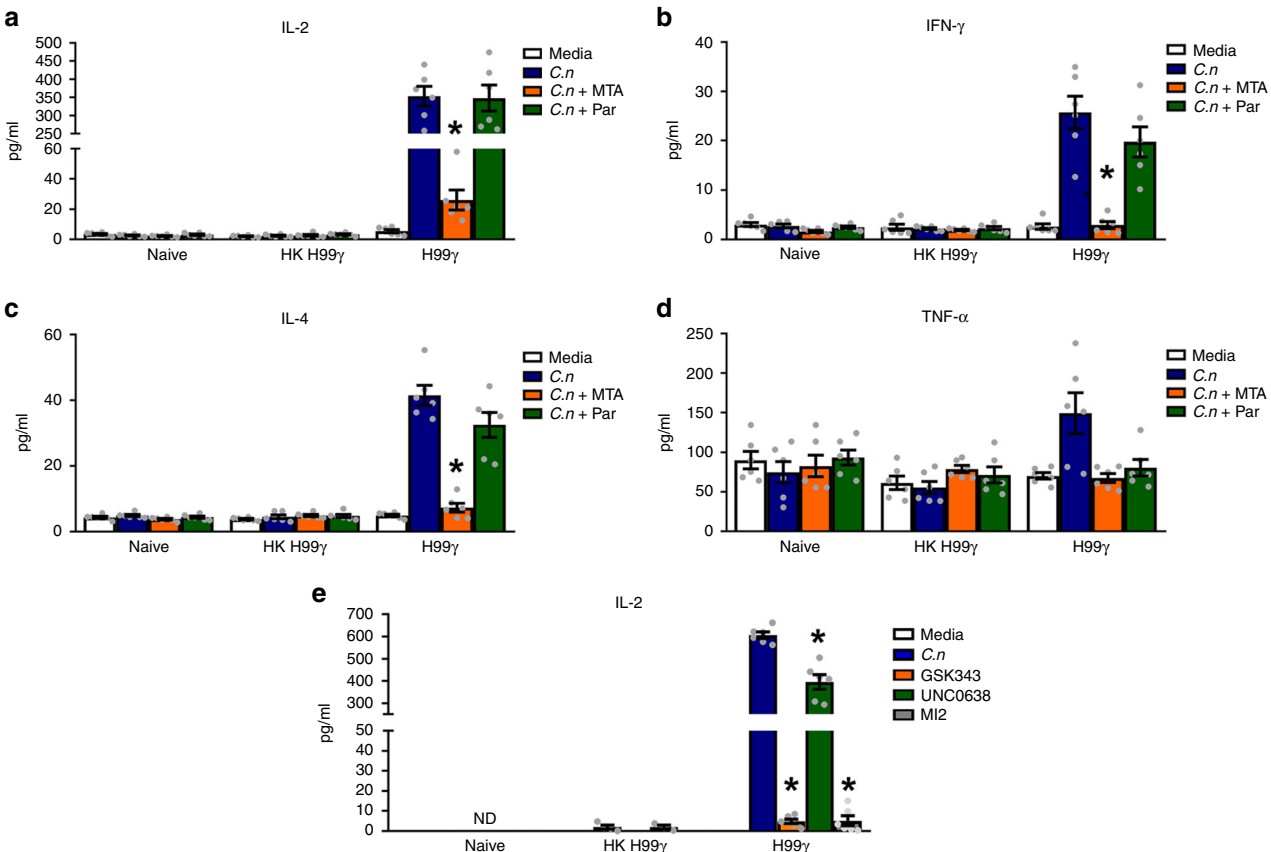

**Fig. 6** DC cytokine recall responses in the presence of histone modification inhibitors. **a** BALB/c mice were immunized via intranasal inhalation with $10^4$ CFU of *C. neoformans* strain H99γ or HKH99γ in 50 μl of sterile PBS. **b**–**d** After 70 days, DCs were isolated from the spleens of naïve, HKH99γ-immunized or H99γ-immunized mice after depletion of CD3⁺ and F4/80⁺ cells. Isolated DCs were cultured in media alone or in media containing live *cna1Δ* ± MTA (methyltransferase inhibitor) or pargyline (demethylase inhibitor) for 24 h and cytokine levels from the supernatants were analyzed. **e** Isolated DCs were also cultured in media alone or in media containing live *cna1Δ* ± GSK343 (EZH2 inhibitor that trimethylates H3K27), UNC0638 (G9a inhibitor that methylates H3K9) or MI-2 (MLL inhibitor that trimethylates H3K4) for 24 h and IL-2 production in culture analyzed. Results are cumulative of three experiments utilizing pooled DCs from 20 mice per group. Bars indicate the mean ± standard error of the mean (SEM). Asterisks indicate where significant differences were observed compared to DCs cultured in media containing live *cna1Δ* (*$p < 0.05$), one-way ANOVA with the Tukey's multiple comparison test

putative metabolic reprogramming of DCs isolated from H99γ immunized mice will certainly need further concentrated analysis to understand its complexity. Overall, DCs from protectively immunized mice were observed to have a significant *C. neoformans*-specific cytokine recall response.

**Histone modifications drive DC innate memory.** Studies have shown that enhanced inflammatory responses by monocytes are associated with changes to their epigenetic programming by histone modifications[36–39]. Certain histone modifications, such as trimethylation of histone 3 lysine 4 (H3K4me3), are associated with open chromatin structure which is permissive to transcription of specific genes whereas other modifications, such as tri-methylation of histone 3 lysine 27 (H3K27me3), are associated with closed chromatin structure, which is repressive to transcription[40]. To assess the role of histone modifications in the enhanced recall responses in DCs from H99γ immunized mice, DCs were isolated as described above from spleens of naïve, HKH99γ immunized or H99γ immunized mice. Isolated DCs were cultured in media alone or in media containing live *cna1Δ* ± MTA, a histone methyltransferase inhibitor, or pargyline, a histone demethylase inhibitor. Treatment with pargyline had no

effect on the levels of IL-2, IFN-γ, and IL-4 cytokine production in culture supernatants following culture of DCs isolated from H99γ immunized mice with *Cryptococcus* (Fig. 6a–c). Treatment with both MTA and pargyline did decrease TNF-α levels back to unstimulated levels, however the difference was not significant (Fig. 6d). Conversely, treatment with MTA resulted in a significant reduction of IL-2, IFN-γ, and IL-4 cytokines produced by DCs isolated from H99γ immunized mice and cultured with *cna1Δ* (Fig. 6a–c), suggesting that the addition of methyl groups to histone tails is important in DC enhanced inflammatory responses. As MTA is a pan methyltransferase inhibitor, we next sought to investigate the role of specific methyltransferases. Isolated splenic DCs were cultured in media alone or in media containing live *cna1Δ* ± GSK343, an inhibitor of the enzyme EZH2, which trimethylates H3K27, UNC0638, an inhibitor of G9a, which methylates H3K9, or MI-2, an inhibitor of the MLL complex, which trimethylates H3K4. IL-2 production in culture supernatants was measured as an indication of the DC cytokine recall response. Treatment with UNC0638 led to a significant reduction in IL-2 production by DCs from protectively immunized mice exposed to *cna1Δ* (Fig. 6e). However, GSK343 and MI-2 treatment resulted in a near complete abrogation of IL-2 production by DCs from protectively immunized mice, reducing

it to levels observed in DCs cultured in media alone (Fig. 6e). These results suggest that enhanced *Cryptococcus*-specific responses exhibited by the IFN-γ primed DCs in vivo are associated with histone modifications to specific amino acids that appear to have disparate effects on cytokine production.

## Discussion

We used an experimental fungal vaccine model to show that vaccine strategies can be designed to elicit pathogen-specific innate memory-like responses by DCs. Mice protectively immunized using *C. neoformans* strain H99γ or non-protectively immunized with HKH99γ yeast displayed distinct DC polarization phenotypes during challenge with WT *C. neoformans*. Mice non-protectively immunized with HKH99γ showed significantly increased gene expression of M2 macrophage polarization associated markers FIZZ1 on days 1 and day 3 post-challenge and CD206 on day 3 post-challenge, whereas mice protectively immunized with H99γ demonstrated significantly increased expression of M1 macrophage activation associated markers NOS2 and CXCL9. These data align with our previous results regarding macrophage polarization phenotypes during protective and non-protective amnestic responses against *C. neoformans*[21]. Additionally, the expression of transcripts for CXCL9 and CXCL10, markers recently associated with trained immunity[41], were significantly increased in DCs derived from H99γ infected mice or H99γ immunized mice challenged with WT *C. neoformans* strain H99 further suggesting that DCs from H99γ immunized mice are driven to acquire innate memory cell activity.

Recent studies indicate that immunological memory is not restricted to the lymphocyte compartment system. Netea *et. al.* proposed the term trained immunity to describe innate cell memory-like responses[32]. Mice vaccinated against tuberculosis with *Mycobacterium bovis* Bacillus Calmette Guerin (BCG) show some protection to *C. albicans* and *Schistosoma mansoni* in a T cell independent manner[32]. Monocytes trained with β-glucans, allowed to rest, and re-stimulated days later produce significantly higher levels of the cytokines TNF-α and IL-6 compared to the un-trained cells[37]. We observed similar results when we stimulated DCs isolated from protectively immunized mice with the temperature sensitive *cna1Δ* derived from *C. neoformans* strain H99. Strikingly, our results were obtained using DCs isolated from protectively immunized mice that have cleared the immunization strain several weeks prior to challenge. The DCs from the protectively immunized mice produced significantly higher levels of IL-2, IFN-γ, IL-4, and TNF-α compared to cultures of DCs derived from naïve or HKH99γ immunized mice following secondary challenge. Although the protectively immunized mice were inoculated with H99γ via intranasal inhalation, it appears that the protective phenotype was also conferred onto splenic DCs. We show evidence that *C. neoformans* strain H99γ disseminates to the spleen likely resulting in activation of splenic DCs and subsequent acquisition of the trained phenotype. Altogether, it appears the impact of immunization with *C. neoformans* strain H99γ via intranasal immunization is not limited to pulmonary tissues.

Our study demonstrates that DCs isolated from protectively immunized mice express a memory-like behavior different from that of DCs isolated from non-protectively immunized mice. However, we observed some additional differences between outcomes of innate training in our findings in DCs and those initially described by Netea in the BCG vaccination model with monocytes and macrophages. Principally, we observed significantly higher levels of IL-2, IFN-γ, and IL-4 cytokine production in cultures of DCs isolated from H99γ immunized mice exposed to

*Cryptococcus*, but not LPS, *C. albicans* or *S. aureus*, compared to cultures of DCs from naïve or non-protectively immunized mice cultured in a similar manner. These results suggest that vaccine strategies can be designed to elicit pathogen-specific innate memory responses. Identification of the structural components of *C. neoformans* responsible for the induction of the memory phenotype in DCs will be important for the development of a more clinically applicable vaccine strategy to induce a combined adaptive and innate memory immune response against cryptococcosis. Such structural components may include chitin deacetylases or various mannoproteins shown to induce protective immune responses against cryptococcosis[42,43] or perhaps cryptococcal β-glucans similar those implicated in the induction of trained immunity and protection against reinfection with *C. albicans*[37]. In contrast, *Cryptococcus* capsular material is likely not to serve as a good candidate as cryptococcal polysaccharides have profound immune suppressive effects[44] and polysaccharide-only vaccines generally do not induce strong or long-lasting immune responses associated with immunological memory.

Enhanced inflammatory responses by monocytes are associated with changes to their epigenetic programming due to histone modifications[36–39]. Treating monocytes with the histone methyltransferase inhibitor MTA during training led to a significant reduction of TNF-α during the recall response, whereas treatment with the histone demethylase inhibitor pargyline had no effect on TNF-α[37]. Treating DCs from protectively immunized mice with MTA significantly reduced the levels of IL-2, IFN-γ, and IL-4 produced during the recall experiment. In line with the observations by Quintin et al.[37], we observed no difference in cytokine levels when the DCs were treated with pargyline. Results of our cytokine recall analysis involving treatment with the more specific methyltransferase inhibitors GSK343, MI-2, and to a lesser extent, UNC0638 suggests that the enhanced inflammatory responses produced by DCs from protectively immunized mice are due to changes in H3K4me3 and H3K27me3. These modifications have been shown to be important in DC responses to sepsis[45], β-glucan monocyte training[37], as well as in macrophage polarization[46]. Patients that recover from sepsis can display an immunoparalysis phenomenon where they have significant deficiencies in their immune responses and have an inability to clear a primary infection or develop a new secondary infection[45]. Using a mouse model of sepsis, it was shown that the DCs become more tolerogenic and fail to produce IL-12 when re-stimulated. This chronic suppression of IL-12 in the DCs was linked to epigenetic changes of H3K4me3 and H3K27me3 at IL-12 promoters repressing the production of this cytokine. While this model is driven by immune suppression and ours by activation, both of our findings indicate that DCs have the potential to develop innate memory-like responses. Moreover, our results suggest that vaccine and immunotherapeutic protocols may be designed to program innate cells, like DCs, to predominantly respond to antigen exposure in a protective or suppressive manner. Understanding the mechanisms underlying innate immune memory has the potential to have a significant clinical impact such as facilitating the design of vaccines that augment adaptive and innate memory responses against diseases particularly problematic to immune compromised individuals. Also, therapies can be designed that reverse immune paralysis such as that induced after sepsis, enhance innate cell responses to cancer, or reprogram cells to be less inflammatory thereby reducing the impact of several autoimmune diseases[17,33].

Epigenetic modifications can be used by cells to control gene expression, are especially important for long-term maintenance of acquired transcriptional states and are heritable. In the immune system, histone modifications are required for T cell memory, tolerance, monocyte differentiation and macrophage polarization.

Changes in DNA methylation also influence the transcriptional activity of cells and may contribute to reprogramming cells to have enhanced anti-microbial activity. Although performed with a limited sample size, studies by Verma et al. suggest that alterations in DNA methylation patterns in response to BCG vaccination enhance anti-mycobacterial macrophage responses[47]. The promoter of IFN-γ remained in a reduced methylated state 8 months post-BCG vaccination in individuals showing enhanced anti-mycobacterial responses. IFN-γ is established as critical to the induction of protective immune responses against cryptococcosis[25,27]. Thus, vaccination with H99γ may have resulted in changes in DNA methylation patterns that contribute to the trained immune phenotype observed herein; however, additional analyses will need to evaluate this possibility. Nonetheless, we demonstrated that DCs can be trained and this training appears to be dependent on epigenetic modifications. Further understanding of how epigenetic modifications control DC polarization, activation and memory-like responses can lead to the discovery of drug-able targets and development of therapies to drive DC polarization towards a phenotype more beneficial for alleviating specific disease processes. Furthermore, vaccine formulations may be designed to drive DC polarization toward a phenotype that will, in turn, drive protective cellular responses; even in immune compromised host populations.

Altogether, we utilized a fungal vaccine strategy to show that approaches can be designed to modulate DC polarization and memory-like responses in vivo. DCs isolated from protectively immunized mice exhibit strong IFN-γ responses and enhanced pro-inflammatory cytokine responses upon subsequent challenge which is indicative of a memory response. Whole transcriptome analysis demonstrated that protective immunization induces DCs that are primed to more effectively respond to subsequent challenge in a pathogen-specific manner weeks following the initial stimulus. Treating DCs with methyltransferase inhibitors abrogated the recall response in DCs from protectively immunized mice showing a critical role for histone modifications in DC biology. These findings reveal aspects of DC biology that can influence how vaccines and/or immune therapies are designed to induce innate cell memory-like immune responses.

## Methods

**Mice.** Female BALB/c ($H$-$2^d$) mice, aged 5–6 weeks, were purchased from the National Cancer Institute/Charles River Laboratories and were housed at The University of Texas at San Antonio Small Animal Laboratory Vivarium. All animal experiments were conducted following NIH guidelines for housing and care of laboratory animals and in accordance with all relevant protocols and ethical regulations for animal testing and research approved by the Institutional Animal Care and Use Committee (protocol number MU021) of the University of Texas at San Antonio.

**Strains and media.** *C. neoformans* strains H99 (serotype A, mating type α) H99γ (serotype A, mating type α, an interferon-gamma producing strain derived from *C. neoformans* H99[25]), and H99 calcineurin mutant (*cna1Δ*, a kind gift from Dr. Joseph Heitman, Duke University) were recovered from 15% glycerol stocks stored at −80 °C and maintained on yeast-extract-peptone-dextrose (YPD) agar (1% yeast extract, 2% peptone, 2% dextrose, and 2% Bacto agar). *Candida albicans* strain SC5314 was maintained on YPD media. *Staphylococcus aureus* strain UAMS-1 was maintained on *Luria*-Bertani (LB) agar. Yeast cells were grown for 15–17 h at 30 °C with shaking in YPD broth (Becton Dickinson and Company, Sparks, MD), collected by centrifugation ($2500 \times g$), washed three times with sterile phosphate-buffered saline (PBS), and viable yeast quantified by trypan blue dye exclusion in a hemacytometer. *S. aureus* was grown for 12 h at 37 °C with shaking in LB broth (Fisher Scientific, Waltham, MA), collected by centrifugation ($2500 \times g$) and washed three times with sterile PBS. Yeast cells were heat-killed at 65 °C for 30 min. *S. aureus* was heat-killed at 95 °C for 1 h. All cultures were confirmed dead by culturing for growth on LB or YPD agar.

**Pulmonary infections.** Pulmonary *C. neoformans* infections were initiated by nasal inhalation[21,26,27]. BALB/c mice were anesthetized with 2% isoflurane using a rodent anesthesia device (Eagle Eye Anesthesia, Jacksonville, FL) and then given a yeast inoculum of $1 \times 10^4$ colony forming units (CFU) of *C. neoformans* strains H99 or H99γ in 50 μl of sterile PBS pipetted directly into one nostril for the primary inoculation. For secondary infection studies, anesthetized BALB/c mice received a yeast inoculum of $1 \times 10^4$ CFU of either *C. neoformans* strain H99γ or heat-killed *C. neoformans* strain H99γ (HKH99γ) yeasts in 50 μl of sterile PBS and were allowed 70 days to resolve the infection. Subsequently, the immunized mice received a second experimental pulmonary inoculation with $1 \times 10^4$ CFU of wild-type *C. neoformans* strain H99 in 50 μl of sterile PBS. The inocula used for primary inoculation, immunizations, and challenge were verified by quantitative culture on YPD agar. The mice were fed ad libitum and were monitored by inspection twice daily. Mice were euthanized at specific time points post-inoculation by $CO_2$ inhalation followed by cervical dislocation, and lung tissues were excised using aseptic technique.

**Pulmonary leukocyte isolation.** Lungs were excised at specific time points post-inoculation and enzymatically digested at 37 °C for 30 min in 10 ml of digestion buffer (RPMI 1640 and 1 mg/ml of collagenase type IV) with intermittent stomacher homogenization. The digested tissues were then successively passed through sterile nylon filters of various pore sizes (70 and 40 μm) (BD Biosciences, San Jose, CA) to enrich for leukocytes and washed with sterile HBSS. Erythrocytes were lysed by incubation in $NH_4Cl$ buffer (0.859% $NH_4Cl$, 0.1% $KHCO_3$, 0.0372% $Na_2EDTA$ [pH 7.4]; Sigma-Aldrich, St. Louis, MO) for 3 min on ice followed by the addition of a two-fold excess of PBS. The cells were then collected by centrifugation ($1000 \times g$), resuspended in sterile PBS + 2% heat-inactivated fetal bovine serum (FACS buffer) and enumerated in a hemacytometer using trypan blue dye exclusion.

**DC enrichment.** Leukocytes were isolated from lungs and spleens as described above. Leukocyte populations were double depleted of $CD3^+$ cells by positive selection using α-CD3ε labeled magnetic beads (Cat. No 13-094-973, Miltenyi Biotec, Auburn, CA), and macrophages by positive selection using biotinylated α-F4/80 (Cat. No 13-4801-85, clone BM8, Affymetrix eBioscience Inc. San Diego, CA) and α-biotin magnetic beads (Cat. No 130-090-485, Miltenyi Biotec) according to the manufacturer's recommendations. DCs were isolated by CD11c (Cat. No 130-108-338, Miltenyi Biotec) positive selection according to the manufacturer's recommendations and purity assessed by flow cytometry analysis to be >95% pure and contain <1.5% TCRα/β+ cells.

**L-Lactate measurement.** DCs were isolated from the lungs of immunized mice as described above. After DCs enrichment, cells were seeded in a 96-well plate at a density of $5 \times 10^5$ cells/well; in triplicate in RPMI complete media ± *C. neoformans* calcineurin mutant (*cna1Δ*; $5 \times 10^5$ cells/well), for 24 h at 37 °C in 5% $CO_2$. After 24 h the supernatant was collected and samples centrifuged at 14,000 g for 30 min with a 10 kDa MWCO spin filter (MRCPRT010, Millipore) to remove lactate dehydrogenase. The L-lactate concentration in the supernatant was colorimetrically determined by an absorbance measurement at 450 nm using an L-lactate assay kit (MAK065, Sigma).

**Flow cytometry.** Standard methodology was employed for the direct immunofluorescence of dendritic cells[27,48]. Briefly, in 96-well U-bottom plates, 100 μl containing $1 \times 10^6$ cells in PBS were incubated with yellow Zombie viability dye (1:1000 dilution, Cat. No 423104, Biolegend, San Diego, CA) for 15 min. at room temperature followed by washing in FACS buffer. Cells were then incubated with Fc block (1:500 dilution, Cat. No 553142, clone 2.4G2, BD Biosciences) diluted in FACS buffer for 5 min to block nonspecific binding of antibodies to cellular Fc receptors. Cells ($1 \times 10^6$ cells) were incubated with fluorochrome-conjugated antibodies targeting CD45-BV711 (1:200 dilution, Cat. No 103147, clone 30-F11, Biolegend), CD11b-APC (1:200 dilution, Cat. No 553312, clone M1/70, BD Biosciences), CD11c-PerCP (1.5:100 dilution, Cat. No 45-914-80, clone N418, eBioscience), CD24-FITC (1:250 dilution, Cat. No 561777, clone M1/69, BD Biosciences), CD64-PeCy7 (1:100 dilution, Cat. No 139313, clone X54-5/7.1, Biolegend), FcεR1α-PE (1:200, Cat. No 12-5898-81, clone MAR-1, eBioscience), Ly6C-BV421 (1:200 dilution, Cat. No 562727, clone AL-21, BD Horizon), I-A/I-E-BV605 (1:100 dilution, Cat. No 563413, clone M5/114.15.2, BD Horizon), CD103-BV786 (1:100 dilution, Cat No, 564322, clone M290, BD Biosciences), CD172α (1:100 dilution, Cat No. 740071, clone P84, BD Biosciences), and TCRα/β-PE (1:200 dilution, Cat. No 553172, clone H57-597, BD Biosciences) in various combinations to allow for multi-staining. The cells were then incubated for 30 min at 4 °C. Cells were washed three times with FACS buffer and fixed in 200 μl of 2% ultrapure formaldehyde (Polysciences, Warrington, PA) diluted in FACS buffer (fixation buffer). Fluorescence minus one (FMO) controls or cells incubated with either FACS buffer alone or single fluorochrome-conjugated Abs were used to determine positive staining and spillover/compensation calculations, and background fluorescence determined with FlowJo v.10 Software (FlowJo, LLC, Ashland, OR). Raw data was collected with a Cell Analyzer LSRII (BD Biosciences) using BD FACS-Diva v8.0 software at the Cell Analysis Core of the UTSA and compensation and data analyses were performed using FlowJo v.10 Software. Cells were first gated for lymphocytes (SSC-A vs FSC-A), and for singlets (FSC-H vs. FSC-A). The singlets gate was further analyzed for the uptake of live/dead yellow stain to

determine live vs. dead cells. From live cells, cells were gated on CD45$^+$ cell expression. For data analyses, 100,000 events (cells) were evaluated from a predominantly leukocyte population identified by back gating from CD45$^+$ stained cells. The absolute number of total DCs was quantified by multiplying the total number of cells observed by hemacytometer counting by the percentage of CD45$^+$ cells determined by flow cytometry. The absolute number of CD103$^+$ conventional DCs (CD11c$^+$/CD24$^+$/CD11b$^-$/CD103$^+$), CD11b$^+$ conventional DCs (CD11c$^+$/CD24$^+$/CD11b$^+$/CD172α$^+$), and CD11b$^+$ monocyte-derived DCs (CD11c$^+$/CD11b$^+$/CD24$^-$/CD64$^+$/FcεR1α$^+$/Ly6C$^+$/I-A/I-E$^+$) determined by multiplying the percentage of each gated population by the total number of CD45$^+$ cells.

**Intracellular cytokine staining and imaging flow cytometry**. Dendritic cells were isolated as described above. DCs were then incubated with *C. neoformans cna1*Δ at a 1:1 ratio at 37 °C in 5% $CO_2$ in RPMI without phenol red plus 10% heat-inactivated fetal bovine serum (Life Technologies, Carlsbad, CA) for 2 h. Golgi plug (1:100 dilution, Brefeldin A, Cat. No 51-2301KZ, BD Biosciences) was added according to manufacturer's recommendations and incubated for an additional 4 h. (6 h total). Cells were washed with PBS and stained with yellow Zombie viability dye (1:1000 dilution, Cat. No 423104, Biolegend, San Diego, CA) in PBS at room temperature in the dark for 15 mins. Cells were then washed with FACS buffer and incubated with Fc block (BD Biosciences) diluted in FACS buffer for 5 min. Cells were stained for surface markers CD45, CD11c and CD24, (Supplementary Fig. 6) as described above and incubated at 4 °C for 30 mins. Cells were washed and fixed with Fix/Permeabilization buffer (BD Biosciences) for 20 mins. Subsequently, cells were washed with Perm/wash buffer (BD Biosciences) and stained with antibodies for cytokines IFN-γ-APC (1:400 dilution; Cat. No 17-7311-81, clone XMG1.2, eBioscience), IL-2-PE (1:400 dilution; Cat. No 561061, clone JES6-5H4, BD Biosciences), TNF-α-PE (1:400 dilution; Cat. No 12-7321-81, clone MP6-XT22, eBioscience) and IL-4-APC (1:400 dilution; Cat. No 17-7041-81, clone 11B11, eBioscience) in different wells for 30 min at 4 °C. Cells were washed with Perm/ wash buffer and fixed with 2% ultra-pure formaldehyde (Polysciences). Samples were processed in triplicate using an ImageStreamX ISX-MKII Imaging Flow Cytometer (Luminex Corporation, Seattle, WA) at the Cell Analysis Core at the UTSA. Analysis was performed using IDEAS® v6.2 software (Luminex Corporation) after 100,000 cells were collected.

**DC gene expression**. Total RNA was isolated from purified DCs using TRIzol reagent (Invitrogen, Carlsbad, CA) and then DNase (Qiagen, Germantown, MD) treated to remove possible traces of contaminating DNA according to the manufacturer's instructions. Total RNA was subsequently recovered using the Qiagen RNeasy kit. cDNA was synthesized from 1 μg total RNA using oligo(dT) primer and reagents supplied in the SuperScript III RT kit (Invitrogen) according to the manufacturer's instructions. The cDNA was used as a template for real-time PCR analysis using the TaqMan gene expression assay (Life Technologies) according to the manufacturer's instructions. All real-time PCR reactions were performed using the 7300 real-time PCR system (Life Technologies). For each real-time PCR reaction, a master mix was prepared on ice with TaqMan gene expression assays specific for NOS2, IFN-γ, IL-17A, CXCL9, CXCL10, Ym1, FIZZ1, Arg1, IL-4, IL-13, and CD206 (Life Technologies). TaqMan rodent GAPDH (Life Technologies) was used as an internal control. The thermal cycling parameters contained an initial denaturing cycle of 95 °C for 10 min followed by 40 cycles of 95 °C for 15 s and 60 °C for 60 s. Results of the real-time PCR data were derived using the comparative $C_t$ method to detect relative gene expression as previously described[21].

**RNA purification and sequencing**. Total RNA was isolated from purified DCs using TRIzol reagent (Invitrogen) and then DNase (Qiagen) treated to remove possible traces of contaminating DNA according to the manufacturer's instructions. Total RNA was subsequently recovered using the Qiagen RNeasy kit. RNA integrity and concentration was assessed via Bioanalyzer using the Agilent RNA 6000 Nano Kit according to the manufacturer's recommendations (Agilent, Santa Clara, CA). Minimum acceptable RNA integrity number (RIN) was set at 7 for use in RNA-sequencing studies. Library preparation was performed using Illumina® TruSeq® RNA Sample Preparation Kit v2 (Illumina, San Diego, CA) and subsequently sequenced on an Illumina Hiseq® 2500 System (Illumina) per manufacturer's instructions. Library preparation and sequencing were performed by UT Southwestern Medical Center Genomics and Microarray Core Facility, Dallas, TX.

**Gene expression analysis**. Data was normalized and differential gene expression was determined using a method that incorporates an internal-standard based approach of normalization and an associative t-test to minimize false positive determinations at the UT Southwestern Medical Center Genomics and Microarray Core Facility, Dallas TX[49,50]. Genes exhibiting normalized expression values 20 times the standard deviation of the statistically defined background were considered expressed. Genes differentially expressed ≥2 fold passed the standard t-test significance level of $p < 0.05$ and passed an associative t-test threshold ($p < 1/$ (number of genes in array expressed above noise) ~1/1000) to eliminate false positive determinations.

**GO, pathway, and network analysis**. GO analysis was performed using the DAVID functional analysis tool[51]. The Bonferroni, Benjamini, and FDR were used for multiple test correction. Functional pathway and network analyses of differentially expressed genes were performed using Ingenuity Pathway Analysis (IPA) (Qiagen). The Ingenuity Knowledge Base, a repository of biological interactions, was used as a reference set. The functional analysis module in IPA was used to identify over-represented molecular and cellular functions of differentially expressed genes. The probability that each biological function assigned to the data set was due to chance alone was estimated, and a false discovery rate (FDR) < 0.05 was used to correct for multiple comparisons. Over-represented canonical signaling and metabolic pathways in the input data were determined based on two parameters: (1) The ratio of the number of molecules from the focus gene set that map to a given pathway divided by the total number of molecules that map to the canonical pathway, and (2) a P-value calculated by Fisher's exact test that determines the probability that the association between the focus loci and the canonical pathway is explained by chance alone. Network analysis used focus genes as seeds to infer de novo interaction networks. Direct interactions between focus loci and other molecules were inferred based on experimentally observed relationships supported by at least one reference from the literature. Additional molecules from the Ingenuity Knowledge Base were added to the network to fill or join smaller networks. The network score was based on the hypergeometric distribution and calculated with the right-tailed Fisher's exact test. A higher score indicates a lower probability of finding the observed number of focus molecules in a given network by chance.

**Cytokine recall assays**. BALB/c mice received a pulmonary immunization with $10^4$ CFU of H99γ or HKH99γ yeast and were allowed 70 days to resolve the infection. The mice were then sacrificed and spleens removed. Spleens were also extracted from uninfected, non-immunized mice to serve as an additional control for our studies. DCs were isolated from the spleen as described above and separately cultured ($5 \times 10^5$/well; in duplicate) within individual wells of a 96-well tissue culture plate in RPMI complete media alone, media ± *C. neoformans* calcineurin mutant (*cna1*Δ; $5 \times 10^5$/well), LPS (Sigma-Aldrich; 0.3 μg/ml), heat-killed *S. aureus* ($5 \times 10^5$/well), or heat-killed *C. albicans* ($5 \times 10^5$/well) for 24 h. Additionally, DCs were cultured ($5 \times 10^5$/well; in triplicate) within individual wells of a 96-well tissue culture plate in RPMI complete media alone, media ± DMFM or DMSO, or RPMI complete media containing live *cna1*Δ ($5 \times 10^5$/well) ±1 mM MTA (methyltransferase inhibitor), 3 μM pargyline (demethylase inhibitor) 4.2 μM GSK343 (EZH2 inhibitor that trimethylates H3K27), 500 mM UNC0638 (G9a inhibitor that methylates H3K9) or 12 μM MI-2 (MLL inhibitor that trimethylates H3K4) all from Cayman Chemicals (Ann Arbor, MI) for 24 h. Supernatants were removed and treated with protease inhibitor (Thermo Fisher Scientific, Waltham, MA), then frozen at −80 °C until use. Cytokine levels were analyzed by Bio-Plex Pro Mouse Cytokine Th1-Th2 8-plex Assay (BioRad, Hercules, CA) or by an IL-2 ELISA (Affymetrix, Santa Clara, CA).

**Statistical analysis**. The unpaired Student's t-test (two-tailed) was used to analyze pulmonary cell populations and cytokine/chemokine data using GraphPad Prism version 5.00 for Windows (GraphPad Prism Software, San Diego CA). For multiple comparisons, a one-way ANOVA with the Tukey's multiple comparison test was performed. Significant differences were defined as $P < 0.05$.

**Reporting summary**. Further information on research design is available in the Nature Research Reporting Summary linked to this article.

## Data availability

The RNA-Seq results presented in this manuscript have been submitted to the Sequence Read Archive under accession number SRP120338. The authors declare that all data supporting the findings of this study are available within the article and its supplementary information files or from the corresponding author upon reasonable request.

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

## Acknowledgements

These studies were supported by research grant 2RO1AI071752 from the National Institute of Allergy and Infectious Diseases (NIAID) of the National Institutes of Health (NIH) (F.L.W. Jr.), the UTSA Microsoft President's Endowed Professorship (F.L.W. Jr.), and grant GM060655 from the National Institute of General Medical Sciences of the NIH (A.C.). The funders had no role in study design, data collection and analysis, decision to publish, or preparation of the manuscript.

## Author contributions

C.R.H.: conceptualization, formal analysis, investigation, methodology, writing–original draft, writing–review and editing. C.M.H.W.: formal analysis, investigation, methodology, writing–review and editing. N.C.L.: formal analysis, investigation, methodology, writing–review and editing, visualization. A.C.: formal analysis, investigation, methodology, writing–review and editing. H.C.: formal analysis; visualization. K.L.W.: formal analysis, investigation, methodology, writing–review and editing. Y.W.: conceptualization, formal analysis, investigation, writing–review and editing. F.L.W. Jr.: conceptualization, formal analysis, investigation, methodology, project administration, resources, supervision, writing–original draft, writing–review and editing

## Additional information

**Competing interests:** The authors declare no competing interests.

