## [Peer Review File · Nature Communications]

Reviewers' comments:

Reviewer #1 (Remarks to the Author):

The present study by Hole and colleagues investigates the capacity of dendritic cells (DCs) to develop memory-like characteristics (also termed trained immunity). The induction of trained immunity has been recently showed to take place in monocytes and macrophages, while similar memory properties have been reported in lymphoid innate cells such as NK-cells and innate lymphoid cells. However, nothing was known until now whether DCs can also develop these properties, and the current study addresses a very important question. Moreover, the question whether DCs have innate immune memory characteristics is also crucial due to their importance for the interaction between the innate and adaptive immunity.

The study is relevant, the experiments are well performed, and the manuscript is clearly written. I have only a few comments and suggestions.

1. Can the authors speculate which are the structural components of *Cryptococcus* who are responsible for the induction of the memory phenotype in DCs?
2. Metabolic changes have been reported to accompany trained immunity phenotype in monocytes and macrophages, especially a shift towards aerobic glycolysis. This can be easily assessed as release of lactate, the end product of glycolysis. Is there also more lactate released by the 'trained' DCs?
3. It has been recently shown that not only histone modifications, but also DNA methylation, accompany trained immunity phenotype after BCG vaccination (Verma et al, *Sci Rep.* 2017 Sep 26; 7(1):12305). Can the authors discuss whether *Cryptococcus* vaccination may also induce changes in DNA methylation?
4. It would be useful for the readers if the authors would discuss in more depth the clinical implications of their discovery.

Reviewer #2 (Remarks to the Author):

1 Answer to the Nat Comm questions:

What are the major claims of the paper?

The previous work from the authors demonstrated that strain H99y induces protection in a cryptococcosis model caused by *C. neoformans*. The experiments conducted in this work have used this strain to demonstrate the polarization to DC1, as evidenced by the quantification of important markers. Gene expression / global transcriptome studies of DC demonstrated important genes induced by strain H99y. Next, the authors demonstrated that histone modifications are important in this DC polarization.

Are they novel and will they be of interest to others in the community and the wider field? yes

If the conclusions are not original, it would be helpful if you could provide relevant references. Is the work convincing, and if not, what further evidence would be required to strengthen the conclusions? See comments below

On a more subjective note, do you feel that the paper will influence thinking in the field? yes

2. Major comments:

The results from this study clearly points to the increased production of proinflammatory mediators by pulmonary DC population after mice vaccination with H99y, leading to the DC1 polarization, which is altered by some histone inhibitors. The main question from this reviewer is about some features in the murine model that would be useful to explore. Considering that the work focused on the fungus *C. neoformans* and also considering the most severe disease caused by the fungus (cryptococcal meningoencephalitis), the authors should verify the role of the pulmonary DC polarization also in the transmigration of the fungus from the lungs to the central nervous system

(CNS). Was the fungal burden in the brain influenced by the H99y vaccine? Do the inflammatory mediators produced by the DC influence evolution of the disease? The H99 strain is studied worldwide and lung inoculation thereof is known to be followed by the involvement of CNS. Although previous studies of the group already demonstrate that the vaccine strain acts on the host's response against the fungus, results on the evolution of the disease would be important to add value to the work, since it would be possible to associate / correlate important mediators in DC1 with parameters such as behavior, weight, lung fungal load and ability of the fungus to translocate to the CNS. Qualitative (eg, PCA) or statistical analysis to associate or even correlate the increased expression of DC1 markers with disease evolution in the animal model would certainly enrich the work and would provide new perspectives on the vaccine development.

3. Minor comments:

Fungal burden should be expressed as CFU/g of organ.

Authors stated at the first section of the results that " ...the infection was almost resolved (day 14)..." but the fungal burden in the lungs is still higher than 10³CFU/mL. This reviewer thinks the infection is not yet resolved, but it can be considered controlled.

Fig 2, please state that UC is unchallenged group in the fig. legend.

In section 2 of the results, the authors stated "...and correlating with the spike in Arg1 transcript levels seen within DCs...". The authors should cite in the materials and methods the statistical test used for the correlation analysis.

Reviewer #3 (Remarks to the Author):

The authors describe an interesting difference in the phenotype of myeloid inflammatory infiltrates in response to engineered *Cryptococcus*. Their previous work had characterized various macrophage phenotypes. Here they go further showing that the infiltrates include CD11c⁺ cells that they characterize as DC. In the second part of the paper, they isolate similar cells from the spleens of previously vaccinated animals and show that they exhibit a range of recall responses, some cross-reactive. Exposure to inhibitors of histone methylation inhibits some of these *in vitro* responses but not in a very specific manner.

MAJOR POINTS

1) the authors propose a distinction between DC1 and DC2 in a parallel fashion to M1 and M2. While it is clear that this 'polarization' phenomenon is distinct from the ontogenetic classification cDC1 / cDC2, it is not clear from their experiments how the polarized DC1/2 differ from their previous reports of M1/M2 macrophages described in the same system. Looking back at the previous work: 'pulmonary leukocytes were isolated from lung tissues by enzymatic digestion on days 1, 3, 7, and 14 post-inoculation and macrophages were enriched for by positive selection of CD11b⁺ cells'. In the current paper, DC are defined as CD11b⁺ CD11c⁺ cells. Were these 'DC' not included in the preparations that were previously described as macrophages?

In the present study, the authors removed macrophages with antibodies to F4/80 and then enriched for DC with CD11c. It is essential that the resulting preparations are described in more detail in order to know exactly what cells are being described. For example – are CD11b⁺ cDC2 still included? Or are the cells mainly monocyte-derived DC? How do the current experiments relate to the enriched macrophage populations described previously? While the current experiments represent a refinement of the previous data, it is not clear to what extent the data really define a novel phenomenon in DC biology, as opposed to a description of monocyte-derived inflammatory cells.

2) The gene expression analysis in Figure 3 is from pulmonary leukocytes isolated from immunized mice but the *in vitro* 'recall' responses are from splenic DC. There was no attempt to explain a mechanism that would confer a protective phenotype on splenic DC, when the original exposure was to the lung. I am concerned that the phenomenon observed in the spleen reflects a state of adaptive memory mediated by T cells with a bystander effect upon DC. The DC preparations were not characterized in detail and the reader needs to be reassured that there was no carry over of T

cells into the in vitro assays, that might account for the increase in cytokine recall responses. When the authors considered explanations for the observed DC recall responses, the data appeared to have features of both trained immunity and antigen specific responses. There was no clear explanation of why some responses had features of trained immunity while others showed features antigen specificity.

3) The use of histone methylation inhibitors is interesting but the results are difficult to interpret. In the final sentence of the results it is stated that MTA, UNC0638, GSK343 (EZH2 inhibitor that interferes with H3K27me3) and MI-2 (MLL inhibitor that interferes with H3K4me3) all suppress IL-2 production and that this indicates that 'specific histone modifications' are responsible for the DC priming phenomenon. It is hard to visualize this as a specific mechanism, given that all of these inhibitors have the same effect on the assay reported.

Overall, the reviewers expressed that our studies were relevant, well performed, the manuscript clearly written, and that the results would influence thinking in the field. Additionally, the reviewers provided some very insightful comments that allowed us to significantly improve the clarity and impact of the study. Consequently, we are thankful and appreciative of the time and effort that the reviewers placed in reviewing our original manuscript. We carefully considered the reviewers critique of our original submission and included new data as well as point-by-point responses to address the reviewers concerns. Overall, we believe that the manuscript is significantly improved as a result and is ready for publication in *Nature Communications*.

Reviewer 1:

Critique:

1. Can the authors speculate which are the structural components of *Cryptococcus* who are responsible for the induction of the memory phenotype in DCs?

Response:

1. We have carefully considered this comment and, consequently, revised the “Discussion” section of the revised manuscript to include some speculation regarding *Cryptococcus* structural components that may or may not be responsible for the induction of the memory phenotype in DCs (lines 305 - 315). Previous studies to elucidate *Cryptococcus* components that may be involved in the induction of trained immunity or protective immune responses against fungi or *C. neoformans* guided our discussion on this topic.

Critique:

2. Metabolic changes have been reported to accompany trained immunity phenotype in monocytes and macrophages, especially a shift towards aerobic glycolysis. This can be easily assessed as release of lactate, the end product of glycolysis. Is there also more lactate released by the ‘trained’ DCs?

Response:

2. We performed studies to determine if a shift towards aerobic glycolysis occurred in DCs from H99 γ immunized mice by accessing changes in the release of lactate release as suggested by Reviewer 1. We observed no significant difference in lactate release between DCs from HKH99 γ compared to H99 γ immunized mice following culture in the presence of the *C. neoformans cna1* Δ for 24h (lines 220 – 227). We note in the revised manuscript that this does not rule out a role for metabolic changes as a contributing factor to facilitate the innate memory phenomenon observed herein. Nevertheless, this line of questioning is worthy of focused scrutiny that we are in the process of pursuing but believe is beyond the scope of this manuscript.

Critique:

3. It has been recently shown that not only histone modifications, but also DNA methylation, accompany trained immunity phenotype after BCG vaccination (Verma et al, Sci Rep.

2017 Sep 26;7(1):12305). Can the authors discuss whether *Cryptococcus* vaccination may also induce changes in DNA methylation?

Response:

3. We are thankful to the reviewer for bringing this study to our attention. We revised the “Discussion” section of the manuscript to include a brief discussion of this study and whether vaccination with *C. neoformans* strain H99 γ may have resulted in changes in DNA methylation patterns that contribute to the trained immune phenotype observed herein (lines 348 - 357).

Critique:

4. It would be useful for the readers if the authors would discuss in more depth the clinical implications of their discovery.

Response:

4. We agree with the reviewer. Consequently, we revised the current manuscript to include some additional discussion regarding the clinical implications of our discovery (lines 338 - 344).

Reviewer 2:

Critique:

1. The results from this study clearly points to the increased production of proinflammatory mediators by pulmonary DC population after mice vaccination with H99 γ , leading to the DC1 polarization, which is altered by some histone inhibitors. The main question from this reviewer is about some features in the murine model that would be useful to explore. Considering that the work focused on the fungus *C. neoformans* and also considering the most severe disease caused by the fungus (cryptococcal meningoencephalitis), the authors should verify the role of the pulmonary DC polarization also in the transmigration of the fungus from the lungs to the central nervous system (CNS). Was the fungal burden in the brain influenced by the H99 γ vaccine? Do the inflammatory mediators produced by the DC influence evolution of the disease? The H99 strain is studied worldwide and lung inoculation thereof is known to be followed by the involvement of CNS. Although previous studies of the group already demonstrate that the vaccine strain acts on the host's response against the fungus, results on the evolution of the disease would be important to add value to the work, since it would be possible to associate/correlate important mediators in DC1 with parameters such as behavior, weight, lung fungal load and ability of the fungus to translocate to the CNS.

Response:

1. The revised manuscript now includes data and further consideration of the impact of DC polarization on dissemination of *Cryptococcus* to brain and splenic tissues in H99 γ

compared to H99 infected mice (lines 65 – 82). We include results showing weight loss, pulmonary fungal burden and yeast burden in brain and splenic tissues of H99 γ compared to H99 infected mice as indicative of dissemination to the CNS and elsewhere (Figure 1). Overall, these data show that pulmonary infection with H99 γ results in DC1 polarization and reduction of pulmonary fungal burden and dissemination to the CNS and spleen. We believe that the inclusion of these results will allow the reader to observe the impact of H99 γ infection on DC polarization and amelioration of cryptococcosis.

Critique:

2. Fungal burden should be expressed as CFU/g of organ.

Response:

2. The revised manuscript now expresses fungal burden in the lung, brain, and spleen as CFU/g of organ (Figure 1).

Critique:

3. Authors stated at the first section of the results that “...the infection was almost resolved (day 14)...” but the fungal burden in the lungs is still higher than 103CFU/mL. This reviewer thinks the infection is not yet resolved, but it can be considered controlled.

Response:

3. We agree with the reviewer. Therefore, we revised the manuscript to refer to the fungal burden at day 14 post-infection in H99 γ infected mice as controlled (line 89).

Critique:

4. Fig 2, please state that UC is unchallenged group in the fig. legend.

Response:

4. We have revised the figure legend to state that UC is the unchallenged group (Fig 3, line 566 in revised manuscript)

Critique:

5. In section 2 of the results, the authors stated "...and correlating with the spike in Arg1 transcript levels seen within DCs...". The authors should cite in the materials and methods the statistical test used for the correlation analysis.

Response:

5. Our intent was to bring attention to a parallel increase in NOS2 and Arg1 transcript levels within DCs isolated from H99 γ infected mice on day 3 post challenge. We did not perform any correlation analysis as we were attempting to bring attention to increases in transcript

levels occurring within the same population of cells. We have revised the manuscript to better clarify our intent (lines 132 – 136) and apologize for the lack of clarity.

Reviewer 3:

Critique:

1. The authors propose a distinction between DC1 and DC2 in a parallel fashion to M1 and M2. While it is clear that this ‘polarization’ phenomenon is distinct from the ontogenetic classification cDC1 / cDC2, it is not clear from their experiments how the polarized DC1/2 differ from their previous reports of M1/M2 macrophages described in the same system. Looking back at the previous work: ‘pulmonary leukocytes were isolated from lung tissues by enzymatic digestion on days 1, 3, 7, and 14 post-inoculation and macrophages were enriched for by positive selection of CD11b⁺ cells’. In the current paper, DC are defined as CD11b⁺ CD11c⁺ cells. Were these ‘DC’ not included in the preparations that were previously described as macrophages?

Response:

1. The reviewer poses an excellent question. The answer to the question “Were these ‘DC’ not included in the preparation that were previously described as macrophages?” is, in short, not completely. Our initial characterization of macrophage polarization, which has since been updated in subsequent publications, included selection of macrophages by positive selection of CD11b⁺ cells and undoubtedly included some DCs. However, the data provided within our revised manuscript includes a much broader characterization of the DC subsets in the lungs of H99 compared to H99 γ infected mice. In essence, we observe robust recruitment of CD103⁺ conventional (c) DCs (CD11c⁺/CD24⁺/CD11b⁻/CD103⁺) and CD11b⁺cDCs (CD11c⁺/CD24⁺/CD11b⁺/CD172 α ⁺) in the lung (Figure 1). Thus, the populations that we are describing herein are limited to DCs alone and include both CD11b⁺ and CD11b⁻ DCs.

Critique:

2. In the present study, the authors removed macrophages with antibodies to F4/80 and then enriched for DC with CD11c. It is essential that the resulting preparations are described in more detail in order to know exactly what cells are being described. For example – are CD11b⁺ cDC2 still included? Or are the cells mainly monocyte-derived DC? How do the current experiments relate to the enriched macrophage populations described previously? While the current experiments represent a refinement of the previous data, it is not clear to what extent the data really define a novel phenomenon in DC biology, as opposed to a description of monocyte-derived inflammatory cells.

Response:

2. The studies presented in Figure 1 of the revised manuscript includes a more detailed characterization of the DC populations. Briefly, our analysis shows that our analyses are

specific to DCs and not macrophage populations and point to a phenomenon that is specific to DCs.

Critique:

3. The gene expression analysis in Figure 3 is from pulmonary leukocytes isolated from immunized mice but the in vitro ‘recall’ responses are from splenic DC. There was no attempt to explain a mechanism that would confer a protective phenotype on splenic DC, when the original exposure was to the lung. I am concerned that the phenomenon observed in the spleen reflects a state of adaptive memory mediated by T cells with a bystander effect upon DC. The DC preparations were not characterized in detail and the reader needs to be reassured that there was no carry over of T cells into the in vitro assays, that might account for the increase in cytokine recall responses. When the authors considered explanations for the observed DC recall responses, the data appeared to have features of both trained immunity and antigen specific responses. There was no clear explanation of why some responses had features of trained immunity while others showed features antigen specificity.

Response:

3. The reviewer expresses an important concern. Figure 1 of the revised manuscript provides evidence that *C. neoformans* strain H99 γ disseminate to the spleen likely resulting in activation of splenic DCs and subsequent DC acquisition of the innate memory phenotype. This is a likely explanation for how the protective phenotype is conferred onto splenic DCs of H99 γ infected mice (discussed in lines 290 – 295 of the revised manuscript). Also, our DC isolation strategy involved performing a double depletion of CD3⁺ T cells prior to depletion of F4/80⁺ cells (macrophages) and isolation of DCs. Flow cytometry analysis for T cell contamination showed less than 1.5% T cells in the final preparation, an amount akin to background staining. Also, we used imaging flow cytometry to validate the production of IFN- γ , TNF- α , IL-2, and IL-4 by DCs in our cytokine recall assays (Supplemental figure 3 of the revised manuscript). Thus, we believe that our data demonstrates that the phenomenon that we observe is from the DCs.

Critique:

4. The use of histone methylation inhibitors is interesting but the results are difficult to interpret. In the final sentence of the results it is stated that MTA, UNC0638, GSK343 (EZH2 inhibitor that interferes with H3K27me3) and MI-2 (MLL inhibitor that interferes with H3K4me3) all suppress IL-2 production and that this indicates that ‘specific histone modifications’ are responsible for the DC priming phenomenon. It is hard to visualize this as a specific mechanism, given that all of these inhibitors have the same effect on the assay reported.

Response:

4. Figure 6 of our manuscript shows that treatment with the demethylase inhibitor pargyline has no significant effect on cytokine production following culture of DCs isolated from

H99 γ immunized mice with *Cryptococcus*. Additionally, treatment with UNC0638, an inhibitor of G9a that methylates H3K9, did not have as dramatic an impact on cytokine production as GSK343 and MI-2 which completely abrogated cytokine production. Thus, we believe that the inhibitors did not have the same effect towards inhibiting cytokine production by DCs of H99 γ vaccinated mice cultured with *Cryptococcus*. Nonetheless, we also note, in the revised manuscript, that other epigenetic modifications such as DNA methylation may influence the transcriptional activity of cells and contribute to reprogramming cells to have enhanced anti-microbial activity (line 348 - 357).

Reviewers' comments:

Reviewer #1 (Remarks to the Author):

The authors responded appropriately to my suggestions.

Reviewer #2 (Remarks to the Author):

The authors improved the manuscript, added new data and answered all the questions from this reviewer. The article may now, according to the position of this reviewer, be accepted for publication.

Daniel A. Santos

Reviewer #3 (Remarks to the Author):

The authors remain unclear about the relative contributions of cDC1, cDC2 and monocyte-derived DC to their findings. As shown in the revised Figure 1, there is a differential and time-dependent skewing of CD103+ cDC1 and CD11b+ cDC2 recruitment to the lung, in response to H99 or H99gamma. Monocyte-derived DC can be distinguished from cDC2 in this organ, as indicated by papers cited by the authors, but this distinction was not made. This means that all of the results that follow, may be due to the differential recruitment of cDC1, cDC2 and monocyte-derived DC lineages. Therefore, the 'memory' that was observed has an important explanation in terms of relative contributions of DC subsets that has been overlooked. This would accord well with recent descriptions of innate memory that invoke an explanation in hematopoiesis.

The problem that remains, is that even though the authors pay passing recognition to cDC1 and cDC2 in Figure 1, they then ignore this vital observation and revert to an uncharacterised 'CD11c+ DC' population for all of the subsequent work. From the results in Figure 1, it would be strongly anticipated that all of the differential gene expression and function that follows, are explained by different composition of the infiltrate, in terms of cDC1 and cDC2 and an unknown quantity of monocyte-derived cells. The authors did not provide information on the phenotypic composition of the cells they subsequently define as 'CD11c+ DC' as requested in the critique.

Furthermore, the unorthodox and confusing terms 'DC1' and 'DC2' are retained in the manuscript. The papers cited to justify this (9 and 10) do not, in my opinion, support the statement in the manuscript: 'Since recent studies showed that DCs are capable of polarizing to DC1 or DC2 phenotypes'. For example, Connor et al (citation 9) carefully define the origin of their DC populations: 'We found that parasite material is taken up by two distinct DC populations in draining lymph nodes: a mostly CD11c^{int}MHC class II (MHCII)^{hi}CD11b⁺Ly6C² dermal DC population and a CD11c^{hi}MHCII^{int}CD11b⁺Ly6C⁺ monocyte-derived DC population'. Also Tjota and Sperling (citation 10), carefully describe in their paper a distinction between cDC2 and mo-DCs when they state: Development of Type-2 immune responses during helminth infections, allergic airway inflammation, or atopic dermatitis has been most frequently associated with both CD11b⁺ conventional DCs (cDCs) and CD11b⁺ monocyte-derived DCs (moDCs).

In order to reach a state of the art description of the phenomena described in relation to H99 and H99gamma I would expect the current manuscript to pay the same level of attention to the phenotypic composition of the CD11c DC infiltrate as do these papers published 4 years ago in J Immunol and Curr Op in Immunol. Neither paper cited introduces the terms DC1 and DC2 and these terms are best omitted. In the absence of any information on the phenotypic composition of the DC infiltrates, the authors should use non-inferential terms like 'control' and 'gamma-induced' to describe their observations.

We are thankful and appreciative of the time and effort that the reviewers placed in reviewing our revised manuscript. Overall, reviewers 1 and 2 expressed that we responded appropriately to their critiques and were favorable to acceptance of the manuscript for publication. Nevertheless, there remained matters that required additional clarification and we welcome the opportunity to further improve the manuscript. We included new data as well as point-by-point responses to address the remaining concerns. Overall, we believe that the manuscript is improved as a result and is ready for publication in *Nature Communications*.

Reviewer 1:

Critique:

The authors responded appropriately to my suggestions.

Reviewer 2:

Critique:

The authors improved the manuscript, added new data and answered all the questions from this reviewer. The article may now, according to the position of this reviewer, be accepted for publication.

Reviewer 3:

Critique:

The authors remain unclear about the relative contributions of cDC1, cDC2 and monocyte-derived DC to their findings. As shown in the revised Figure 1, there is a differential and time-dependent skewing of CD103⁺ cDC1 and CD11b⁺ cDC2 recruitment to the lung, in response to H99 or H99gamma. Monocyte-derived DC can be distinguished from cDC2 in this organ, as indicated by papers cited by the authors, but this distinction was not made. This means that all of the results that follow, may be due to the differential recruitment of cDC1, cDC2 and monocyte-derived DC lineages. Therefore, the ‘memory’ that was observed has an important explanation in terms of relative contributions of DC subsets that has been overlooked. This would accord well with recent descriptions of innate memory that invoke an explanation in hematopoiesis.

Response:

We appreciate the reviewer’s concern that the CD11b⁺ cDC2 population identified in our studies may include monocyte-derived DCs that may be a major contributor to the ‘memory’ DC responses observed in our studies. We agree that it’s important to address this matter and, therefore, included new data in our revised manuscript showing that the enriched DC population demonstrated in our studies as having memory responses are absent of monocyte-derived DCs. Specifically, we demonstrated that less than 1% of the DC population derived from the spleens of *C. neoformans* strain H99γ and HKH99γ immunized mice and utilized in our cytokine recall

assays were monocyte-derived DCs (CD11c⁺/CD11b⁺/CD24⁻/CD64⁺/FceR1a⁺/Ly6C⁺/I-A/I-E⁺). We utilized a panel of markers typically used to identify monocyte-derived DCs to make this determination (Tjota, M. Y. *et al.* Distinct dendritic cell subsets actively induce Th2 polarization. *Current opinion in immunology* **31**, 44-50, doi:10.1016/j.coi.2014.09.006 and Schlitzer, A. *et al.* IRF4 transcription factor-dependent CD11b⁺ dendritic cells in human and mouse control mucosal IL-17 cytokine responses. *Immunity* **38**, 970-983, doi:10.1016/j.immuni.2013.04.011 and Langlet, C. *et al.* CD64 expression distinguishes monocyte-derived and conventional dendritic cells and reveals their distinct role during intramuscular immunization. *Journal of immunology* **188**, 1751-1760, doi:10.4049/jimmunol.1102744). We do not find our results surprising considering that the DCs used in the cytokine-recall studies were derived from immunized but unchallenged mice and monocyte-derived DCs are typically not observed in tissues under non-inflammatory conditions (Tjota, M. Y. *et al.* Distinct dendritic cell subsets actively induce Th2 polarization. *Current opinion in immunology* **31**, 44-50, doi:10.1016/j.coi.2014.09.006 and Robays, L. J. *et al.* Chemokine receptor CCR2 but not CCR5 or CCR6 mediates the increase in pulmonary dendritic cells during allergic airway inflammation. *Journal of immunology* **178**, 5305-5311). This information is discussed within the Results section (lines 202 – 209) and included as Supplemental Figure 3 of the revised manuscript.

Critique:

Furthermore, the unorthodox and confusing terms ‘DC1’ and ‘DC2’ are retained in the manuscript. The papers cited to justify this (9 and 10) do not, in my opinion, support the statement in the manuscript: ‘Since recent studies showed that DCs are capable of polarizing to DC1 or DC2 phenotypes’. For example, Connor et al (citation 9) carefully define the origin of their DC populations: ‘We found that parasite material is taken up by two distinct DC populations in draining lymph nodes: a mostly CD11c^{int}MHC class II (MHCII)^{hi}CD11b⁺Ly6C² dermal DC population and a CD11c^{hi}MHCII^{int}CD11b⁺Ly6C⁺ monocyte-derived DC population’. Also Tjota and Sperling (citation 10), carefully describe in their paper a distinction between cDC2 and mo-DCs when they state: Development of Type-2 immune responses during helminth infections, allergic airway inflammation, or atopic dermatitis has been most frequently associated with both CD11b⁺ conventional DCs (cDCs) and CD11b⁺ monocyte-derived DCs (moDCs).

Response:

We concur that the terms ‘DC1’ and ‘DC2’ can be confusing; particularly as we also refer to the conventional terms cDC1 and cDC2. Consequently, the current manuscript has been revised to remove the terms DC1 and DC2.

Critique:

In order to reach a state of the art description of the phenomena described in relation to H99 and H99gamma I would expect the current manuscript to pay the same level of attention to the phenotypic composition of the CD11c DC infiltrate as do these papers published 4 years ago in *J immunol* and *Curr Op in Immunol*. Neither paper cited introduces the terms DC1 and DC2 and these terms are best omitted. In the absence of any information on the phenotypic composition of

the DC infiltrates, the authors should use non-inferential terms like 'control' and 'gamma-induced' to describe their observations.

Response:

We agree with the reviewer and have revised the manuscript to not utilize the terms DC1 and DC2. Additionally, we have included new data within the manuscript to further define the CD11C⁺ DC population utilized in our DC cytokine-recall assays (Supplemental Figure 3) and in the lungs of *C. neoformans* strain H99 γ and HKH99 γ immunized mice at day 1 post-challenge (Supplemental Figure 1). We utilized a broad panel of markers to show that < 1% of the CD11c⁺ cells present in the enriched DC population (lines 202 – 209 Supplemental Figure 3 of the revised manuscript) and in the lungs of immunized mice at day 1 post-challenge (lines 121 – 129 Supplemental Figure 1 of the revised manuscript) are monocyte-derived DCs (i.e., CD11c⁺/CD11b⁺/CD24⁻/CD64⁺/FcεR1a⁺/Ly6C⁺/I-A/I-E⁺ cells).

Additional changes to manuscript:

We have revised the manuscript to include a Data Availability Statement and revised the figures to show individual data points when possible.

REVIEWERS' COMMENTS:

Reviewer #3 (Remarks to the Author):

Thank you for responses to outstanding points raised.